# Towards Better Evaluation of GNN Expressiveness with BREC Dataset

**Yanbo Wang**   **Muhan Zhang**
Institute for Artificial Intelligence, Peking University
`yanxwb202@gmail.com, muhan@pku.edu.cn`

## Abstract

Research on the theoretical expressiveness of Graph Neural Networks (GNNs) has developed rapidly, and many methods have been proposed to enhance the expressiveness. However, most methods do not have a uniform expressiveness measure except for a few that strictly follow the $k$-dimensional Weisfeiler-Lehman ($k$-WL) test hierarchy. Their theoretical analyses are often limited to distinguishing certain families of non-isomorphic graphs, leading to difficulties in quantitatively comparing their expressiveness. In contrast to theoretical analysis, another way to measure expressiveness is by evaluating model performance on certain datasets containing 1-WL-indistinguishable graphs. Previous datasets specifically designed for this purpose, however, face problems with difficulty (any model surpassing 1-WL has nearly 100% accuracy), granularity (models tend to be either 100% correct or near random guess), and scale (only a few essentially different graphs in each dataset). To address these limitations, we propose a new expressiveness dataset, **BREC**, which includes 400 pairs of non-isomorphic graphs carefully selected from four primary categories (Basic, Regular, Extension, and CFI). These graphs have higher difficulty (up to 4-WL-indistinguishable), finer granularity (able to compare models between 1-WL and 3-WL), and a larger scale (400 pairs). Further, we synthetically test 23 models with higher-than-1-WL expressiveness on our BREC dataset. Our experiment gives the first thorough comparison of the expressiveness of those state-of-the-art beyond-1-WL GNN models. We expect this dataset to serve as a benchmark for testing the expressiveness of future GNNs. Our dataset and evaluation code are released at: https://github.com/GraphPKU/BREC.

## 1   Introduction

GNNs have been extensively utilized in bioinformatics, recommender systems, social networks, and others, yielding remarkable outcomes [1–6]. Despite impressive empirical achievements, related investigations have revealed that GNNs exhibit limited abilities to distinguish non-isomorphic graphs, such as regular graphs. In a practical scenario, the inability to recognize structure may cause issues, such as confused representation of a benzene ring (a six-cycle that cannot be recognized). Xu et al. [7], Morris et al. [8] established a connection between the expressiveness of message-passing neural networks (MPNNs) and the WL test for graph isomorphism testing, demonstrating that MPNN's upper bound is 1-WL. Numerous subsequent studies have proposed GNN variants with enhanced expressiveness [9–13].

Given the multitude of models employing different approaches, such as feature injection, adherence to the WL hierarchy, equivariance maintenance, and subgraph extraction, a unified framework that can theoretically compare the expressive power among various variants is highly desirable. In this regard, Maron et al. [14] propose the concept of $k$-order invariant/equivariant graph networks, which unify linear layers while preserving permutation invariance/equivariance. Additionally, Frasca

Submitted to the 37th Conference on Neural Information Processing Systems (NeurIPS 2023) Track on Datasets and Benchmarks. Do not distribute.

et al. [15] unify recent subgraph GNNs and establish that their expressiveness upper bound is 3-WL. Zhang et al. [16] construct a comprehensive expressiveness hierarchy for subgraph GNNs, providing counterexamples for each pairwise distinction. Nonetheless, the magnitude of the gaps remains unknown. Furthermore, there exist methods that are difficult to categorize within the $k$-WL hierarchy. For instance, Papp and Wattenhofer [17] propose four extensions of GNNs, each of which cannot strictly compare with the other. Similarly, Feng et al. [18] propose a GNN that is partially stronger than 3-WL yet fails to distinguish many graphs that are distinguishable by 3-WL. In a different approach, Huang et al. [19] propose evaluating expressiveness by enumerating specific significant substructures, such as 6-cycles. Zhang et al. [20] introduces graph biconnectivity to test expressiveness.

Without a unified theoretical characterization of expressiveness, employing expressiveness datasets for testing proves valuable. Notably, three expressiveness datasets, EXP, CSL, and SR25, have been introduced by Abboud et al. [21], Murphy et al. [22], Balcilar et al. [9] and have found widespread usage in recent studies. However, these datasets exhibit notable limitations. Firstly, they lack sufficient difficulty. The EXP and CSL datasets solely consist of examples where 1-WL fails, and most recent GNN variants have achieved perfect accuracy on these datasets. Secondly, the granularity of these datasets is too coarse, which means that graphs in these datasets are generated using a single method, resulting in a uniform level of discrimination difficulty. Consequently, the performance of GNN variants often falls either at random guessing (completely indistinguishable) or 100% (completely distinguishable), thereby hindering the provision of a nuanced measure of expressiveness. Lastly, these datasets suffer from small sizes, typically comprising only a few substantially different graphs, raising concerns of incomplete measurement.

To overcome the limitations of current expressiveness datasets, we propose a new dataset, BREC, including 400 pairs of non-isomorphic graphs in 4 major categories: Basic graphs, Regular graphs, Extension graphs, and CFI graphs. Compared to previous ones, BREC has a greater difficulty (up to 4-WL-indistinguishable), finer granularity (able to compare models between 1-WL and 3-WL), and larger scale (800 non-isomorphic graphs organized as 400 pairs), addressing the shortcomings.

Due to the increased size and diversity of the dataset, the traditional classification task may not be suitable for training-based evaluation methods which rely on generalization ability. Thus, we propose a novel evaluation procedure based on directly comparing the discrepancies between model outputs to test pure practical expressiveness. Acknowledging the impact of numerical precision owning to tiny differences between graph pairs, we propose reliable paired comparisons building upon a statistical method [23, 24], which offers a precise error bound. Experiments verify that the evaluation procedure aligns well with known theoretical results.

Finally, we comprehensively compared 23 representative beyond-1-WL models on BREC. Our experiments first give a **reliable empirical comparison** of state-of-the-art GNNs' expressiveness. The currently most thorough investigation is a good start for gaining deeper insights into various schemes to enhance GNNs' expressiveness. On BREC, GNN accuracies range from 41.5% to 70.2%, with I$^2$-GNN [19] performing the best. The 70.2% highest accuracy also implies that the dataset is **far from saturation**. We expect BREC can serve as a benchmark for testing future GNNs' expressiveness. We also welcome contributions and suggestions to improve BREC. Our dataset and evaluation code are included in https://github.com/GraphPKU/BREC.

## 2 Limitations of Existing Datasets

**Preliminary.** We utilize the notation $\{\}$ to represent sets and $\{\{\}\}$ to represent multisets. The cardinality of a (multi)set $\mathbb{S}$ is denoted as $|\mathbb{S}|$. The index set is denoted as $[n] = 1, \ldots, n$. A graph is denoted as $\mathcal{G} = (\mathbb{V}(\mathcal{G}), \mathbb{E}(\mathcal{G}))$, where $\mathbb{V}(\mathcal{G})$ represents the set of *nodes* or *vertices* and $\mathbb{E}(\mathcal{G})$ represents the set of *edges*. Without loss of generality, we assume $|\mathbb{V}(\mathcal{G})| = n$ and $\mathbb{V}(\mathcal{G}) = [n]$.

The permutation or reindexing of $\mathcal{G}$ is denoted as $\mathcal{G}^{\pi} = (\mathbb{V}(\mathcal{G}^{\pi}), \mathbb{E}(\mathcal{G}^{\pi}))$ with the permutation function $\pi : [n] \to [n]$, s.t. $(u, v) \in \mathbb{E}(\mathcal{G}) \iff (\pi(u), \pi(v)) \in \mathbb{E}(\mathcal{G}^{\pi})$. Node and edge features are excluded from the definitions for simplicity. Additional discussions about features can be found in Appendix B.

**Graph Isomorphism (GI) Problem.** Two graphs $\mathcal{G}$ and $\mathcal{H}$ are considered isomorphic (denoted as $\mathcal{G} \simeq \mathcal{H}$) if $\exists \phi$(a bijection mapping) :$\mathbb{V}(\mathcal{G}) \to \mathbb{V}(\mathcal{H})$ s.t. $(u, v) \in \mathbb{E}(\mathcal{G})$ iff. $(\phi(u), \phi(v)) \in \mathbb{E}(\mathcal{H})$.

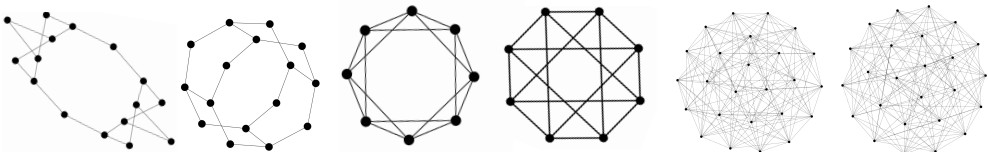

(a) EXP dataset core pair sample  (b) CSL graphs ($m = 10, r = 2$)  (c) SR25 dataset sample

Figure 1: Sample graphs in previous datasets

Table 1: Dataset statistics

| Dataset | # Graphs | # Core graphs[a] | # Nodes | Hardness | Metrics |
|---------|----------|------------------|---------|----------|---------|
| EXP | 1200 | 6 | 33-73 | 1-WL-indistinguishable | 2-way classification |
| CSL | 150 | 10 | 41 | 1-WL-indistinguishable | 10-way classification |
| SR25 | 15 | 15 | 25 | 3-WL-indistinguishable | 15-way classification |
| **BREC** | **800** | **800** | **10-198** | **1-WL to 4-WL-indistinguishable** | **Reliable Paired Comparisons** |

[a] Core graphs represent graphs that actually serve to measure expressiveness.

GI is essential in expressiveness. Only if GNN successfully distinguishes two non-isomorphic graphs can they be assigned different labels. Some researchers [25, 26] indicate the equivalence between GI and function approximation, underscoring the importance of GI. However, we currently do not have polynomial-time algorithms for solving the GI problem. A naive solution involves iterating all $n!$ permutations to test whether such a bijection exists.

**Weisfeiler-Lehman algorithm (WL).** WL is a well-known isomorphism test relying on color refinement [27]. In each iteration, WL assigns a state (or color) to each node by aggregating information from its neighboring nodes' states. This process continues until convergence, resulting in a multiset of node states representing the final graph representation. While WL effectively identifies most non-isomorphic graphs, it may fail in certain simple graphs, leading to the development of extended versions. One such extension is $k$-WL, which treats each $k$-tuple of nodes as a unit for aggregating information. Another slightly different method [28] is also referred to as $k$-WL. To avoid confusion, we follow Morris et al. [8] to call the former $k$-WL and the latter $k$-FWL. Further information can be found in Appendix C.

Given the significance of GI and WL, several expressiveness datasets have been introduced, with the following three being the most frequently utilized. We selected a pair of graphs from each dataset, which are illustrated in Figure 1. Detailed statistics for these datasets are presented in Table 1.

**EXP Dataset.** This dataset comprises 600 pairs of non-isomorphic graphs where the 1-WL test fails. Graphs are generated pair-wised, and each graph comprises two disconnected components. The first component, the "core component," is designed to be non-isomorphic with the other graph's "core component," each satisfying distinct SAT conditions in the two graphs. The second component, referred to as the "planar component," is identical in both graphs and introduces noise into the dataset. However, it is important to note that there are only **three substantially different** core pairs, which can truly evaluate the expressiveness of the models.

Each graph in EXP is labeled 0/1 based on whether its core component satisfies the SAT condition for a binary classification problem. Although EXP addresses the issue of semantic labeling by introducing SAT problem and enhances the dataset's size and complexity by including planar components, the simplicity of core c generation and the insufficient number of different core pairs result in most recent GNNs achieving nearly 100% accuracy on EXP, making it difficult for detailed comparisons.

**CSL Dataset.** This dataset consists of 150 Circulant Skip Links (CSL) graphs, where the 1-WL test fails. A CSL graph is defined as follows: Let $r$ and $m$ be co-prime natural numbers with $r < m - 1$. $\mathcal{G}(m, r) = (\mathbb{V}, \mathbb{E})$ is an undirected 4-regular graph with $\mathbb{V} = [m]$, where the edges form a cycle and include skip links. Specifically, for the cycle, $(j, j+1) \in \mathbb{E}$ for $j \in [m-1]$, and $(m, 1) \in \mathbb{E}$. For the skip links, the sequence is recursively defined as $s_1 = 1$, $s_{i+1} = (s_i + r) \bmod m + 1$, and $(s_i, s_{i+1}) \in \mathbb{E}$ for any $i \in \mathbb{N}$. In CSL, we consider CSL graphs with $m = 41$ and $r = 2, 3, 4, 5, 6, 9, 11, 12, 13, 16$, resulting in 10 distinct CSL graphs. For each distinct CSL graph, we generate 14 corresponding graphs by randomly reindexing the nodes. As a result, the dataset contains a total of 150 graphs.

In CSL, each of the 10 distinct CSL graphs is treated as a separate class, and the task is to train a 10-way classification model. While the dataset allows for the generation of 4-regular graphs with any number of nodes, the final dataset contains only **ten essentially different** regular graphs with the

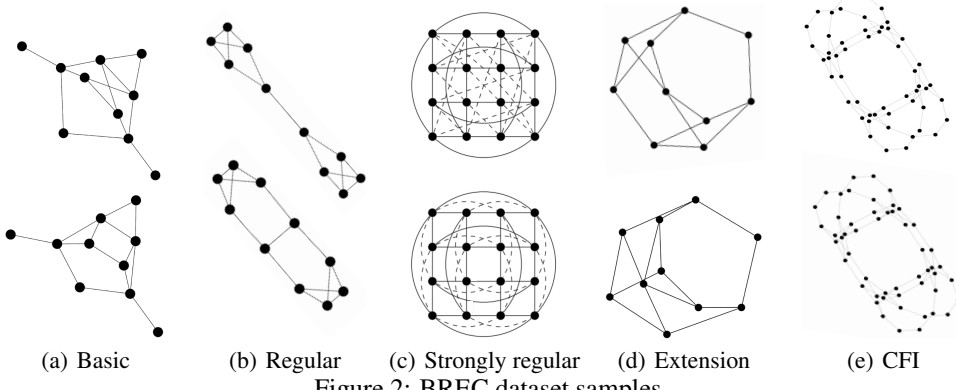

| (a) Basic | (b) Regular | (c) Strongly regular | (d) Extension | (e) CFI |

Figure 2: BREC dataset samples

**same number of nodes and degree**. Due to the nature of regular graphs and their fixed structure, many recent expressive GNN models perform well on this dataset, achieving close to 100% accuracy.

**SR25 Dataset.** It consists of 15 strongly regular graphs (SR) where the 3-WL test fails. Each graph is an SR with 25 nodes and a degree of 12. In these graphs, connected nodes have 5 common neighbors, while non-connected nodes have 6. In practice, SR25 is transformed into a 15-way classification problem for mapping each graph into a different class where the training and test graphs overlap.

Indeed, 3-WL serves as an upper bound for most recent expressive GNNs. Thus most methods only obtain 6.67% (1/15) accuracy. While some models partially surpassing 3-WL easily achieve completely distinguishable (100%) performance [18], since each graph is an SR with the same parameters. This binary outcome can hardly provide a fine-grained expressiveness measure.

**Summary.** These three datasets have limitations regarding difficulty, granularity, and scale. In terms of difficulty, these datasets are all bounded by 3-WL, failing to evaluate models (partly) beyond 3-WL [18, 19]. In terms of granularity, the graphs are generated in one way, and the parameters of the graphs are repetitive, which easily leads to a 0/1 step function of model performance and cannot measure subtle differences between models. In terms of scale, the number of substantially different graphs in the datasets is small, and the test results may be incomplete to reflect expressiveness measurement.

## 3 BREC: A New Dataset for Expressiveness

We propose a new expressiveness dataset, BREC, to address the limitations regarding difficulty, granularity, and scale. It consists of four major categories of graphs: Basic, Regular, Extension, and CFI. Basic graphs include relatively simple 1-WL-indistinguishable graphs. Regular graphs include four types of subcategorized regular graphs. Extension graphs include special graphs that arise when comparing four kinds of GNN extensions [17]. CFI graphs include graphs generated by CFI methods[1] [28] with high difficulty. Some samples are shown in Fig 2.

### 3.1 Dataset Composition

BREC includes 800 non-isomorphic graphs arranged in a pairwise manner to construct 400 pairs, with detailed composition as follows: (For a more detailed generation process, please refer to AppendixK)

**Basic Graphs.** Basic graphs consist of 60 pairs of 10-node graphs. These graphs are collected from an exhaustive search and intentionally designed to be non-regular. Although they are 1-WL-indistinguishable, most can be distinguished by expressive GNN variants. Basic graphs can also be regarded as an augmentation of the EXP dataset, as they both employ non-regular 1-WL-indistinguishable graphs. Nevertheless, Basic graphs offer a greater abundance of instances and more intricate graph patterns. The relatively small size also facilitates visualization and analysis.

**Regular Graphs.** Regular graphs consist of 140 pairs of regular graphs, including 50 pairs of simple regular graphs, 50 pairs of strongly regular graphs, 20 pairs of 4-vertex condition graphs, and 20 pairs

---
[1]CFI is short for Cai-Furer-Immerman algorithm, which can generate counterexample graphs for any k-WL.

of distance regular graphs. Each pair of graphs shares identical parameters. A regular graph refers to a graph where all nodes possess the same degree. Regular graphs are 1-WL-indistinguishable, and some studies delve into the analysis of GNN expressiveness from this perspective [29, 13]. We denote regular graphs without any special properties as simple regular graphs. When exploring more intricate regular graphs, the concept of strongly regular graphs (where 3-WL fails) is often introduced. Strongly regular graphs further require that the number of neighboring nodes shared by any two nodes depends solely on their connectivity. Notable examples of strongly regular graphs include the $4 \times 4$-Rook's graph and the Shrikhande graph (Fig 2(c)). Additionally, the $4 \times 4$-Rook's graph satisfies the 4-vertex condition property, which signifies that the number of connected edges between the common neighbors of any two nodes is solely determined by their connectivity [30]. It is worth mentioning that the diameter of a connected strongly regular graph is always 2 [31]. A more challenging type of graph known as the distance regular graphs [32] is proposed aiming for extending the diameter. Please refer to Appendix A for a more comprehensive exploration of their relationship.

Regular graphs can also as an enriching addition to the CSL and SR25 datasets. By expanding upon the existing subdivisions of regular graphs, this section widens the range of difficulty and raises the upper bound of complexity. Moreover, unlike the previous datasets, regular graphs are not limited to sharing identical parameters for all graphs within each category, greatly enhancing diversity.

**Extension Graphs.** Extension graphs include 100 pairs of graphs inspired by Papp and Wattenhofer [17]. They proposed 4 types of theoretical GNN extensions: $k$-WL hierarchy-based, substructure-counting-based, $k$-hop-subgraph-based, and marking-based methods. The authors reveal that most of them are not strictly comparable. Leveraging the insights from theoretical analysis and some empirically derived findings, we generated 100 pairs of 1-WL-indistinguishable and 3-WL-distinguishable graphs to improve the granularity. Noting that it was not considered in any of the previous datasets.

**CFI Graphs.** CFI graphs consist of 100 pairs of graphs inspired by Cai et al. [28]. They developed a method to generate graphs distinguishable by $k$-WL but not by $(k-1)$-WL for any $k$. We utilized this method to create 100 pairs of graphs spanning up to 4-WL-indistinguishable, even surpassing the current research's upper bounds. Specifically, 60 pairs are solely distinguishable by 3-WL, 20 are solely distinguishable by 4-WL, and 20 are even 4-WL-indistinguishable. Similar to the previously mentioned parts, CFI graphs were not considered in the previous datasets. As the most challenging part, it pushes the upper limit of difficulty even higher. Furthermore, the graph sizes in this section are larger than other parts (up to 198 nodes). This aspect intensifies the challenge of the dataset, demanding a model's ability to process graphs with heterogeneous sizes effectively.

## 3.2 Advantages

**Difficulty.** By utilizing the CFI method, we specifically provide graphs being 4-WL-indistinguishable. Additionally, we include 4-vertex condition graphs and distance regular graphs, which are variants of strongly regular graphs (3-WL-indistinguishable) but pose greater challenges in terms of complexity.

**Granularity.** The different classes of graphs in BREC exhibit varying difficulty levels, each contributing to the dataset in distinct ways. Basic graphs contain fundamental 1-WL-indistinguishable graphs, similar to the EXP dataset, as a starting point for comparison. Regular graphs extend the CSL and SR25 datasets. The major components of regular graphs are simple regular graphs and strongly regular graphs, where 1-WL and 3-WL fail, respectively. Including 4-vertex condition graphs and distance regular graphs further elevates the complexity. Extension graphs bridge the gap between 1-WL and 3-WL, offering a finer-grained comparison for evaluating models beyond 1-WL. CFI graphs span the spectrum of difficulty from 1-WL to 4-WL-indistinguishable. By comprehensive graph composition, BREC explores the boundaries of graph pattern distinguishability.

**Scale.** While previous datasets relied on only tens of different graphs to generate the dataset, BREC utilizes a collection of 800 different graphs. This significant increase in the number of graphs greatly enhances the diversity. The larger graph set in BREC also contributes to a more varied distribution of graph statistics. In contrast, previous datasets such as CSL and SR25 only have the same number of nodes and degrees across all graphs. For detailed statistics of BREC, please refer to Appendix D.

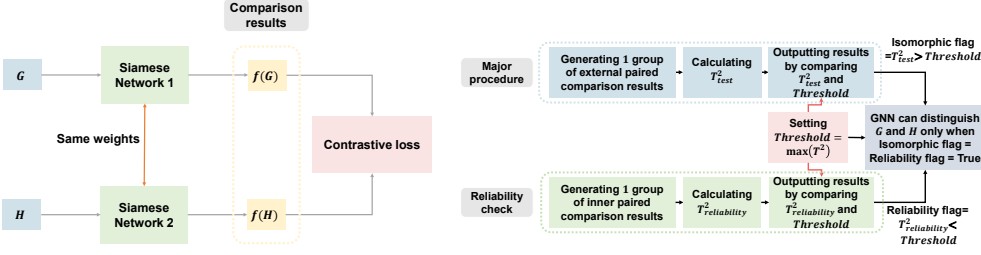

(a) Training Framework                     (b) RPC Pipeline

Figure 3: Evaluation Method

## 4 RPC: A New Evaluation Method

This section introduces a novel training framework and evaluation method for BREC. Unlike previous datasets, BREC departs from the conventional classification setting, where each graph is assigned a label, a classification model is trained, and the accuracy on test graphs serves as the measure of expressiveness. The labeling schemes used in previous datasets like semantic labels based on SAT conditions in EXP, or distinct labels for essentially different graphs in CSL and SR25, do not apply to BREC. There are two primary reasons. First, BREC aims to enrich the diversity of graphs, which precludes using a semantic label tied to SAT conditions, as it would significantly limit the range of possible graphs. Second, assigning a distinct label to each graph in BREC would result in an 800-class classification problem, where performance could be influenced by factors other than expressiveness. Our core idea is to measure models' "separating power" directly. Thus BREC is organized in pairs, where each pair is individually tested to determine whether a GNN can distinguish them. By adopting a pairwise evaluation method, BREC provides a more focused measure of models' expressiveness, aligning to assess distinguishing ability.

Nevertheless, how can we say a pair of graphs is successfully distinguished? Previous researchers tend to set a small threshold (like 1E-4) manually. If the embedding distance between them is larger than the threshold, the GNN is considered can distinguish them. However, this method lacks **reliability** due to numerical precision, especially when graphs vary in size. In order to yield dependable outcomes, we propose an evaluation method measuring both **external difference** and **internal fluctuations**. Furthermore, we introduce a training framework for pairwise data, employing the siamese network design [33] and contrastive loss [34, 35]. The pipeline is depicted in Fig 3(a).

### 4.1 Training Framework

We adhere to the siamese network design [33] to train a model to distinguish each pair of graphs. The central component consists of two identical models that maintain identical parameters. When a pair of graphs is inputted, it produces a corresponding pair of embeddings. Subsequently, the difference between them is assessed using cosine similarity. The loss function is formulated as follows:

$$L(f, \mathcal{G}, \mathcal{H}) = \text{Max}(0, \frac{f(\mathcal{G}) \cdot f(\mathcal{H})}{||f(\mathcal{G})|| \, ||f(\mathcal{H})||} - \gamma), \tag{1}$$

where the GNN model $f : \{\mathcal{G}\} \rightarrow \mathbb{R}^d$, $\mathcal{G}$ and $\mathcal{H}$ are two non-isomorphic graphs, and $\gamma$ is a margin hyperparameter (set to 0 in our experiments). The loss function aims to promote the cosine similarity value lower than $\gamma$, thereby encouraging a greater separation between the two graph embeddings.

The training process **yields several benefits** for the models. Firstly, it enables the GNN to achieve its theoretical expressiveness. The theoretical analysis of GNN expressiveness focuses primarily on the network's structure without imposing any constraints on its parameters, which means we are exploring the expressiveness of **a group of functions**. If a model with particular parameters can distinguish a pair of graphs, the model's design and structure possess sufficient expressiveness. However, it is impractical to iterate all possible parameter combinations to test the real upper bound. Hence, training can **realize searching** in the function space, enabling models to achieve better practical expressiveness. Furthermore, training aids components to **possess specific properties**, such as injectivity and universal approximation, which are vital for attaining theoretical expressiveness. These properties require specific parameter configurations, and randomly initialized parameters may not satisfy these requirements. Moreover, through training, model-distinguishable pairs are **more easily discriminated** from model-indistinguishable pairs, which helps reduce the false negative rate

caused by numerical precision. The difference between their embeddings is further magnified in the pairwise contrastive training process if the model distinguishes them. However, the difference remains unaffected mainly and is only influenced by numerical errors for model-indistinguishable pairs. The training framework is illustrated in Fig 3(a).

## 4.2 Evaluation Method

Recall that our approach involves comparing the outputs on a pair of non-isomorphic graphs. If there exists a notable disparity between them, we consider the GNN to be able to distinguish them. However, determining an appropriate threshold poses a challenge. A large threshold may yield false negatives where the model is expressive enough, but the observed difference falls short of the threshold. Conversely, a small threshold may result in false positives, where the model fails to distinguish the graphs. However, the fluctuating or numerical errors cause the difference to exceed the small threshold.

To address the issue of fluctuating errors, we draw inspiration from Paired Comparisons [23]. It involves comparing two groups of results instead of a single pair. The influence of random errors is mitigated by repeatedly generating results and comparing the two groups of results. Building upon it, we introduce a method called **R**eliable **P**aired **C**omparison (RPC) to verify whether a GNN genuinely produces distinct outputs for a pair of graphs. The pipeline is depicted in Fig 3(b).

RPC consists of two main components: Major procedure and Reliability check. The Major procedure is conducted on a pair of non-isomorphic graphs to measure their dissimilarity. In comparison, the Reliability check is conducted on graph automorphisms to capture internal fluctuations with numerical precision.

**Major procedure.** For two non-isomorphic graphs $\mathcal{G}$ and $\mathcal{H}$, we create $q$ copies of each by randomly reindexing (operate permutation on node indexes, thus generating an isomorphic graph but with different node orders) them. It results in two groups of graphs, where each copy is represented as:

$$\mathcal{G}_i, \ \mathcal{H}_i, \ i \in [q]. \tag{2}$$

Supposing the GNN $f : \{\mathcal{G}\} \to \mathbb{R}^d$, we first calculate $q$ differences utilizing Paired Comparisons.

$$\boldsymbol{d}_i = f(\mathcal{G}_i) - f(\mathcal{H}_i), \ i \in [q]. \tag{3}$$

**Assumption 4.1.** $\boldsymbol{d}_i$ *are independent* $\mathcal{N}(\boldsymbol{\mu}, \boldsymbol{\Sigma})$ *random vectors.*

The above assumption is based on a more basic assumption that $f(\mathcal{G}_i)$, $f(\mathcal{H}_i)$ follow Gaussian distributions, which presumes that random reindexing only introduces Gaussian noise to the result.

The mean difference between two graph embeddings $\boldsymbol{\mu} = \boldsymbol{0}$ implies the GNN cannot distinguish them. Therefore, we can obtain the distinguishing result by conducting an $\alpha$-level Hotelling's T-square test, comparing the hypotheses $H_0 : \boldsymbol{\mu} = \boldsymbol{0}$ against $H_1 : \boldsymbol{\mu} \neq \boldsymbol{0}$. We calculate the $T^2$-statistic for $\boldsymbol{\mu}$ as:

$$T^2 = q(\overline{\boldsymbol{d}} - \boldsymbol{\mu})^T \boldsymbol{S}^{-1}(\overline{\boldsymbol{d}} - \boldsymbol{\mu}), \tag{4}$$

where

$$\overline{\boldsymbol{d}} = \frac{1}{q} \sum_{i=1}^{q} \boldsymbol{d}_i, \ \boldsymbol{S} = \frac{1}{q-1} \sum_{i=1}^{q} (\boldsymbol{d}_i - \overline{\boldsymbol{d}})(\boldsymbol{d}_i - \overline{\boldsymbol{d}})^T. \tag{5}$$

Hotelling's T-square test proves that $T^2$ is distributed as an $\frac{(q-1)d}{q-d} F_{d,q-d}$ random variable, whatever the true $\boldsymbol{\mu}$ and $\boldsymbol{\Sigma}$ [36]. The theorem establishes a connection between the unknown parameter $\boldsymbol{\mu}$ and a definite probability distribution $F_{d,q-d}$, allowing us to confirm the confidence interval of $\boldsymbol{\mu}$ by testing the distribution fit. In order to test the hypothesis $H_0 : \boldsymbol{\mu} = \boldsymbol{0}$, we substitute $\boldsymbol{\mu} = \boldsymbol{0}$ into Equation (4) to obtain $T^2_{\text{test}} = q\overline{\boldsymbol{d}}^T \boldsymbol{S}^{-1} \overline{\boldsymbol{d}}$. Then, for a specific $\alpha$, an $\alpha$-level test of $H_0 : \boldsymbol{\mu} = \boldsymbol{0}$ versus $H_1 : \boldsymbol{\mu} \neq \boldsymbol{0}$ for a population following $\mathcal{N}(\boldsymbol{\mu}, \boldsymbol{\Sigma})$ distribution accepts $H_0$ (the GNN cannot distinguish the pair) if:

$$T^2_{\text{test}} = q\overline{\boldsymbol{d}}^T \boldsymbol{S}^{-1} \overline{\boldsymbol{d}} < \frac{(q-1)d}{(q-d)} F_{d,q-d}(\alpha), \tag{6}$$

where $F_{d,q-d}(\alpha)$ is the upper $(100\alpha)$th percentile of the $F$-distribution $F_{d,q-d}$ [37] with $d$ and $q-d$ degrees of freedom. Similarly, we reject $H_0$ (the GNN can distinguish the pair) if

$$T_{\text{test}}^2 = q\overline{\boldsymbol{d}}^T \boldsymbol{S}^{-1} \overline{\boldsymbol{d}} > \frac{(q-1)d}{(q-d)} F_{d,q-d}(\alpha). \tag{7}$$

**Reliability check.** Although the above test is theoretically valid for evaluating the expressiveness of GNNs, in practice, it is susceptible to computational precision limitations. These limitations can manifest in various scenarios, such as comparing numbers close to zero or inverting a matrix close to zero, making it difficult to rely on the test constantly. We incorporate the Reliability check to monitor abnormal results to address this concern. This step effectively bridges the external difference between two graphs and the internal fluctuations within a single graph.

WLOG, we replace $\mathcal{H}$ by reindexing of $\mathcal{G}$, i.e., $\mathcal{G}^\pi$. Thus, we can obtain the internal fluctuations within $\mathcal{G}$ by comparing it with $\mathcal{G}^\pi$, and the external difference between $\mathcal{G}$ and $\mathcal{H}$ by comparing $\mathcal{G}$ and $\mathcal{H}$. We utilize the same step as Major procedure on $\mathcal{G}$ and $\mathcal{G}^\pi$, calculating the $T^2$-statistics as follows:

$$T_{\text{reliability}}^2 = q\overline{\boldsymbol{d}}^T \boldsymbol{S}^{-1} \overline{\boldsymbol{d}}, \tag{8}$$

$$\text{where } \overline{\boldsymbol{d}} = \frac{1}{q}\sum_{i=1}^{q}\boldsymbol{d}_i, \ \boldsymbol{d}_i = f(\mathcal{G}_i) - f(\mathcal{G}_i^\pi), \ i \in [q], \ \boldsymbol{S} = \frac{1}{q-1}\sum_{i=1}^{q}(\boldsymbol{d}_i - \overline{\boldsymbol{d}})(\boldsymbol{d}_i - \overline{\boldsymbol{d}})^T. \tag{9}$$

Recalling that $\mathcal{G}$ and $\mathcal{G}^\pi$ are isomorphic, the GNN should not distinguish between them, implying that $\boldsymbol{\mu} = \boldsymbol{0}$. Therefore, the test result is considered reliable only if $T_{\text{reliability}}^2 < \frac{(q-1)d}{(q-d)} F_{d,q-d}(\alpha)$. Combining the reliability and distinguishability results, we get the complete RPC (Fig 3) as follows:

For each pair of graphs $\mathcal{G}$ and $\mathcal{H}$, we first calculate the threshold value, denoted as Threshold $= \frac{(q-1)d}{(q-d)} F_{d,q-d}(\alpha)$. Next, we conduct the Major procedure on $\mathcal{G}$ and $\mathcal{H}$ for distinguishability and perform the Reliability check on $\mathcal{G}$ and $\mathcal{G}^\pi$ for Reliability. Only when the $T^2$-statistic from the Major procedure, denoted as $T_{\text{test}}^2$, and the $T^2$-statistic from the Reliability check, denoted as $T_{\text{reliability}}^2$, satisfying $T_{\text{reliability}}^2 < \text{Threshold} < T_{\text{test}}^2$, do we conclude that the GNN can distinguishing $\mathcal{G}$ and $\mathcal{H}$.

We further propose **R**eliable **A**daptive **P**airwise **C**omparison (RAPC), aiming to adaptively adjust the threshold and provide an upper bound for false positive rates. In practice, we use **RPC** due to its less computational time and satisfactory performance. For more about RAPC, please refer to Appendix E.

## 5  Experiment

In this section, we evaluate the expressiveness of 23 representative models using our BREC dataset.

**Model selection.** We evaluate six categories of methods: non-GNN methods, subgraph-based GNNs, $k$-WL-hierarchy-based GNNs, substructure-based GNNs, transformer-based GNNs, and random GNNs. Our primary focus will be on the first three categories. We implement four types of non-GNN baselines based on Papp and Wattenhofer [17], Ying et al. [38], including WL test (3-WL and SPD-WL), counting substructures ($S_3$ and $S_4$), neighborhood up to a certain radius ($N_1$ and $N_2$), and marking ($M_1$). We implemented them by adding additional features during the WL test update or using heterogeneous message passing. It is important to note that they are more theoretically significant than practical since they may require exhaustive enumeration or exact isomorphism encoding of various substructures. We additionally included 16 state-of-the-art GNNs, including NGNN [13], DE+NGNN [29], DS/DSS-GNN [10], SUN [15], SSWL_P [16], GNN-AK [39], KP-GNN [18], I$^2$-GNN [19], PPGN [40], $\delta$-k-LGNN [41], KC-SetGNN [42], GSN [43], DropGNN [44], OSAN [45], and Graphormer [38].

Table 2 presents the primary results. $N_2$ achieves the highest accuracy among non-GNNs, and I$^2$-GNN achieves the highest among GNNs. We detail each method's accuracy on different graphs, showing that it matches theoretical results well. Detailed experiment settings are included in Appendix J.

**Non-GNN baselines.** 3-WL successfully distinguishes all Basic graphs, Extension graphs, simple regular graphs and 60 CFI graphs as expected. $S_3$, $S_4$, $N_1$, and $N_2$ demonstrate excellent performance on small-radius graphs such as Basic, Regular, and Extension graphs. However, due to their limited receptive fields, they struggle to distinguish large-radius graphs like CFI graphs. Noting that the

Table 2: Pair distinguishing accuracies on BREC

| Model | Basic Graphs (60) | | Regular Graphs (140) | | Extension Graphs (100) | | CFI Graphs (100) | | Total (400) | |
|---|---|---|---|---|---|---|---|---|---|---|
| | Number | Accuracy | Number | Accuracy | Number | Accuracy | Number | Accuracy | Number | Accuracy |
| 3-WL | 60 | 100% | 50 | 35.7% | 100 | 100% | 60 | 60.0% | 270 | 67.5% |
| SPD-WL | 16 | 26.7% | 14 | 11.7% | 41 | 41% | 12 | 12% | 83 | 20.8% |
| $S_3$ | 52 | 86.7% | 48 | 34.3% | 5 | 5% | 0 | 0% | 105 | 26.2% |
| $S_4$ | 60 | 100% | 99 | 70.7% | 84 | 84% | 0 | 0% | 243 | 60.8% |
| $N_1$ | 60 | 100% | 99 | 85% | 93 | 93% | 0 | 0% | 252 | 63% |
| $N_2$ | 60 | 100% | 138 | 98.6% | 100 | 100% | 0 | 0% | 298 | 74.5% |
| $M_1$ | 60 | 100% | 50 | 35.7% | 100 | 100% | 41 | 41% | 251 | 62.8% |
| NGNN | 59 | 98.3% | 48 | 34.3% | 59 | 59% | 0 | 0% | 166 | 41.5% |
| DE+NGNN | 60 | 100% | 50 | 35.7% | 100 | 100% | 21 | 21% | 231 | 57.8% |
| DS-GNN | 58 | 96.7% | 48 | 34.3% | 100 | 100% | 16 | 16% | 222 | 55.5% |
| DSS-GNN | 58 | 96.7% | 48 | 34.3% | 100 | 100% | 15 | 15% | 221 | 55.2% |
| SUN | 60 | 100% | 50 | 35.7% | 100 | 100% | 13 | 13% | 223 | 55.8% |
| SSWL_P | 60 | 100% | 50 | 35.7% | 100 | 100% | 38 | 38% | 248 | 62% |
| GNN-AK | 60 | 100% | 50 | 35.7% | 97 | 97% | 15 | 15% | 222 | 55.5% |
| KP-GNN | 60 | 100% | 106 | 75.7% | 98 | 98% | 11 | 11% | 275 | 68.8% |
| I$^2$-GNN | 60 | 100% | 100 | 71.4% | 100 | 100% | 21 | 21% | 281 | 70.2% |
| PPGN | 60 | 100% | 50 | 35.7% | 100 | 100% | 23 | 23% | 233 | 58.2% |
| $\delta$-k-LGNN | 60 | 100% | 50 | 35.7% | 100 | 100% | 6 | 6% | 216 | 54% |
| KC-SetGNN | 60 | 100% | 50 | 35.7% | 100 | 100% | 1 | 1% | 211 | 52.8% |
| GSN | 60 | 100% | 99 | 70.7% | 95 | 95% | 0 | 0% | 254 | 63.5% |
| DropGNN | 52 | 86.7% | 41 | 29.3% | 82 | 82% | 2 | 2% | 177 | 44.2% |
| OSAN | 56 | 93.3% | 8 | 5.7% | 79 | 79% | 5 | 5% | 148 | 37% |
| Graphormer | 16 | 26.7% | 12 | 10% | 41 | 41% | 10 | 10% | 79 | 19.8% |

expressiveness of $S_3$ and $S_4$ is bounded by $N_1$ and $N_2$, respectively, as analyzed by Papp and Wattenhofer [17]. Conversely, $M_1$ is implemented by heterogeneous message passing, which makes it unaffected by large graph diameters, thus maintaining its performance across different graphs. SPD-WL is another 1-WL extension operated on a complete graph with shortest path distances as edge features. It may degrade to 1-WL on low-radius graphs, causing its relatively poor performance.

**Subgraph-based GNNs.** Regarding subgraph-based models, they can generally distinguish almost all Basic graphs, simple regular graphs and Extension graphs. However, an exception lies with NGNN, which performs poorly in Extension graphs due to its simplicial node selection policy and lack of node labeling. Two other exceptions are KP-GNN and I$^2$-GNN, both exhibiting exceptional performance in Regular graphs. KP-GNN can differentiate a substantial number of strongly regular graphs and 4-vertex condition graphs, surpassing the 3-WL partially. And I$^2$-GNN surpasses the limitations of 3-WL partially through its enhanced cycle-counting power. An influential aspect that impacts the performance is the subgraph radius. Approaches incorporating appropriate encoding functions are expected to yield superior performance as the subgraph radius increases. However, in practice, enlarging the radius may result in the smoothness of information, wherein the receptive field expands, encompassing some irrelevant or noisy information. Hence, we treat the subgraph radius as a hyperparameter, fine-tuning it for each model, and present the best results in Table 2. Please refer to Appendix F for further details regarding the radius selection.

When comparing various subgraph GNNs, KP-GNN can discriminate part of the strongly regular graphs by peripheral subgraphs. Additionally, distance encoding in DE+NGNN and I$^2$-GNN enables better discrimination among different hops within a given subgraph radius, enhancing the discriminative ability, particularly in larger subgraph radii. As for DS-GNN, DSS-GNN, GNN-AK, SUN and SSWL_P, they employ similar aggregation schemes with slight variations in their operations. These models exhibit comparable performance, with SSWL_P outperforming others, which aligns with expectations since SSWL_P is more expressive but with the least components.

$k$-**WL hierarchy-based GNNs.** For the $k$-WL-hierarchy-based models, we adopt two implemented approaches: high-order simulation and local-WL simulation. PPGN serves as the representative work for the former, while $\delta$-k-LGNN and KCSet-GNN embody the latter. PPGN aligns its performance with 3-WL across all graphs except for CFI graphs. For CFI graphs with large radii, more WL iterations (layers of GNNs) are required. However, employing many layers may lead to over-smoothing, resulting in a gap between theoretical expectations and actual performance. Nonetheless, PPGN still surpasses most GNNs in CFI graphs due to global $k$-WL's global receptive field. For $\delta$-k-LGNN, we set $k = 2$, while for KCSet-GNN, we set $k = 3, c = 2$ to simulate local 3-WL, adhering to the original configuration. By comparing the output results with relatively small diameters, we observed that local WL matches the performance of general $k$-WL. However, local WL exhibits lower performance for CFI graphs with larger radii due to insufficient receptive fields.

**Substructure-based GNNs** For substructure-based GNNs, we select GSN, which incorporate substructure isomorphism counting as features. The best result obtained for GSN-e is reported when setting $k = 4$. For further exploration of policy and size, please refer to Appendix H.

**Random GNNs** Random GNNs are unsuitable for GI problems since even identical graphs can yield different outcomes due to inherent randomness. However, the RPC can quantify fluctuations in the randomization process, thereby enabling the testing of random GNNs. We test DropGNN and OSAN. For more information regarding the crucial factor of random samples, please refer to Appendix I.

**Transformer-based GNNs** For transformer-based GNNs, we select graphormer, which is anticipated to possess a level of expressiveness comparable to SPD-WL. The experimental results verify that.

## 6 Conclusion and Future Work

This paper proposes a new dataset, BREC, for GNN expressiveness comparison. BREC addresses the limitations of previous datasets, including difficulty, granularity, and scale, by incorporating 400 pairs of diverse graphs in four categories. A new evaluation method is proposed for principled expressiveness evaluation. Finally, a thorough comparison of 23 baselines on BREC is conducted.

Apart from the expressiveness comparison based on GI, there are various other metrics for GNN expressiveness evaluation, such as substructure counting, diameter counting, and biconnectivity checking. However, it's worth noting that these tests are often conducted on datasets not specifically designed for expressiveness [19, 39, 46], which can lead to biased results caused by spurious correlations. In other words, certain methods may struggle to identify a particular substructure, but they can capture another property that correlates with substructures, resulting in false high performance. This problem can be alleviated in BREC because of the difficulty. We reveal the data generation process of BREC in Appendix K, hoping that researchers can utilize them in more tasks. We also hope the test of practical expressiveness will aid researchers in exploring its effects on performance in real datasets and other domains.

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
