## A  Details on Regular Graphs

In this section, we introduce the relationship between four types of regular graphs. The inclusion relations of them are shown in Figure 4, but their difficulty relations and inclusion relations are not consistent.

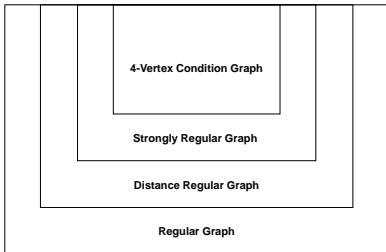

Figure 4: Regular graphs relationship

A graph is deemed a regular graph when all of its vertices possess an identical degree. If a regular graph, with $v$ vertices and degree $k$, satisfies the additional conditions wherein any two adjacent vertices share $\lambda$ common neighbors, and any two non-adjacent vertices share $\mu$ common neighbors, it is categorized as a strongly regular graph. Hence, it can be represented as $\mathrm{srg}(v, k, \lambda, \mu)$, denoting its four associated parameters.

Regular graphs and strongly regular graphs find wide application in expressiveness analysis. The difficulty of strongly regular graphs surpasses that of general regular graphs due to the imposition of additional requirements. Notably, the simplest strongly regular graphs with identical parameters ($\mathrm{srg}(16, 6, 2, 2)$) are exemplified by the Shrikhande graph and the $4 \times 4$-Rook's graph, as depicted in Figure 2(c).

Both 4-vertex condition graphs and distance regular graphs introduce heightened complexities, albeit in opposing directions. A 4-vertex condition graph is a strongly regular graph with an additional property that mandates the determination of the number of edges between the common neighbors of two vertices based on their connectivity. Conversely, distance regular graphs expand upon the definition of strongly regular graphs by specifying that for any two vertices $v$ and $w$, the count of vertices at a distance $j$ from $v$ and at a distance $k$ from $w$ relies solely on $j$, $k$, and the distance between $v$ and $w$. Notably, a distance regular graph with a radius of 2 is equivalent to a strongly regular graph.

The 4-vertex condition graph has yet to be explored in previous research endeavors. Similarly, instances of distance regular graphs are relatively scarce and analyzing them through examples proves to be challenging. To encourage further research in these domains, we have incorporated them into BREC.

## B  Node Features

In this section, we present the concept of node features and edge features in graphs.

We commence by providing the definition of graphs using an adjacency matrix representation. Consider a graph where the node features are represented by a $d_n$-dimensional vector, and the edge features are represented by a $d_e$-dimensional vector. This graph can be denoted as $\mathcal{G} = (\mathbf{V}(\mathcal{G}), \mathbf{E}(\mathcal{G}))$, where $\mathbf{V}(\mathcal{G}) \in \mathbb{R}^{n \times d_n}$ represents the node features, and $\mathbf{E}(\mathcal{G}) \in \mathbb{R}^{n \times n \times (d_e + 1)}$ represents the edge features, with $n$ being the number of nodes in the graph. The adjacency matrix of the graph is denoted as $\mathbf{A}(\mathcal{G}) \in \mathbb{R}^{n \times n} = E(\mathcal{G})_{:,:,(d_e+1)}$, where $\mathbf{A}(\mathcal{G})_{i,j} = 1$ if $(i, j) \in \mathbb{E}(\mathcal{G})$ (i.e., if nodes $i$ and $j$ are connected by an edge), otherwise $A(\mathcal{G})_{i,j} = 0$. The feature of node $i$ is represented by $V(\mathcal{G})_{i,:}$, and the feature of edge $(i, j)$ is represented by $E(\mathcal{G})_{i,j,1:d_e}$. The permutation (or reindexing) of $\mathcal{G}$ is denoted as $\mathcal{G}^\pi = (\mathbf{V}(\mathcal{G}), \mathbf{E}(\mathcal{G}))$ with permutation $\pi : [n] \to [n]$, such that $V(\mathcal{G})_{i,:} = V(\mathcal{G})_{\pi(i),:}$ and $E(\mathcal{G})_{i,j,:} = E(\mathcal{G})_{\pi(i),\pi(j),:}$.

Next, we explore the utilization of features. It is evident that incorporating node features during initialization and edge features during message passing can enhance the performance of GNNs, given

appropriate hyperparameters and training. However, we should consider whether features can truly represent graph structures or provide additional expressiveness. Let us categorize features into two types.

The first type involves fully utilizing the original features, such as distances to other nodes or spectral embeddings. While using these features can aid GNNs in solving Graph Isomorphism (GI) problems, this type of feature requires a dedicated design to effectively utilize them. For instance, if we aim to recognize a 6-cycle in a graph, we can manually identify the cycle and assign distinct features to each node within the cycle. In this way, the GNN can recognize the cycle by aggregating the six distinctive features. However, the injecting strategy influences expressiveness and requires further analysis. Utilizing distance can also enhance expressiveness but also need a suitable design (like subgraph distance encoding and SPD-WL).

The second type entails incorporating additional features, such as manually selected node identifiers. it is important to note that this improvement stems from reduced difficulty rather than increased expressiveness. For instance, given a pair of non-isomorphic graphs with high similarity, we can manually find the components causing the distinguishing difficulty and assign identifiers to help models overcome them. However, this process is generally unavailable in practice.

In summary, we can conclude that features have the potential to introduce expressiveness, but this should be accomplished through model design rather than relying solely on the dataset. In the case of BREC, a dataset created specifically for testing expressiveness, we do not include additional meaningful features. Instead, we employ the same vector for all node features and edge features and adhere to specific model settings to incorporate graph-specific features, such as the distance between nodes in distance encoding based models.

## C  WL Algorithm

This section briefly introduces the WL algorithm and two high-order variants.

The 1-WL algorithm, short for "1-Weisfeiler-Lehman," is an initial version of the WL algorithm. It serves as a graph isomorphism algorithm and can be employed to generate a distinctive label for each graph.

In the 1-WL algorithm, every node in the graph maintains a state or color, which undergoes refinement during each iteration by incorporating information from the states of its neighboring nodes. As the algorithm progresses, the graph representation evolves into a multiset of node states, ultimately converging to a final representation.

To circumvent these examples, researchers have devised a technique to augment each node in the 1-WL test, resulting in the development of the $k$-WL test [47**?** ]. The $k$-dimensional Weisfeiler-Lehman test expands the scope of the test to consider colorings of k-tuples of nodes instead of individual nodes. This extension allows for a more comprehensive analysis of graph structures and assists in overcoming the limitations posed by certain examples.

In addition to the $k$-WL test, Cai et al. [28] proposed an alternative WL test algorithm that also extends to $k$-tuples. This variant is commonly referred to as the $k$-FWL ($k$-folklore-WL) test. The $k$-FWL test differs from the $k$-WL test in terms of how neighbors are defined and the order in which aggregation is performed on tuples and multisets.

There are three notable results associated with these tests:

1  1-WL = 2-WL

2  $k$-WL > $(k-1)$-WL, $(k > 2)$

3  $(k-1)$-FWL = $k$-WL

More details can be found in Sato [48], Huang and Villar [49].

## D  BREC Statistics

Here we give some statistics of the BREC dataset, shown in Figure 5.

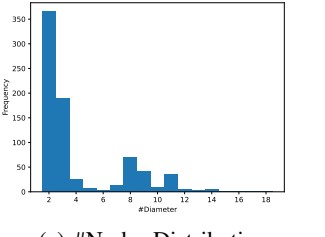
(a) #Nodes Distribution

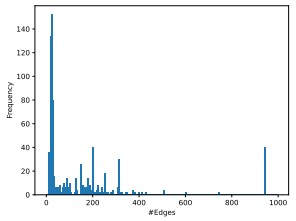
(b) #Edges Distribution

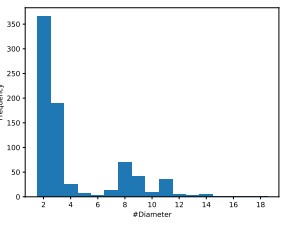
(c) Diameter Distribution

Figure 5: BREC Statistics

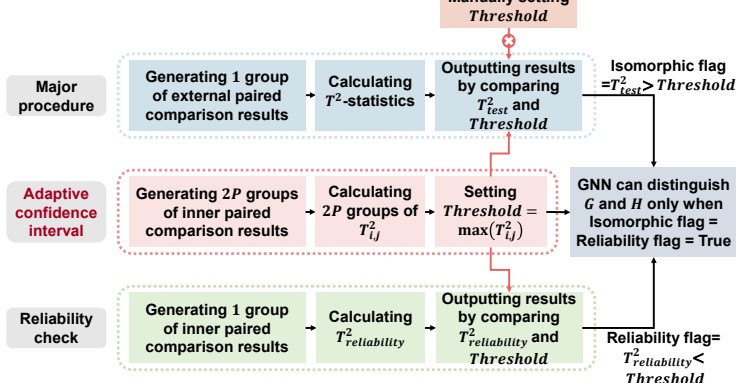

Figure 6: RAPC pipeline.

## E RAPC: a Reliable and Adaptive Evaluation Method

In this section, we propose RAPC with an additional stage called adaptive confidence interval based on RPC. Though RPC performs excellently in experiments with a general theoretical guarantee in reliability, with manually setting $\alpha$. We still want to make the procedure more automated. In addition, we found that the inner fluctuations of each pair, i.e. $T^2_{\text{reliability}}$, vary from pairs. This means some graph outputs are more stable than others, and their threshold can be larger than others. However, it is impossible to manually set the confidence interval ($\alpha$) for all pairs, thus, we propose an adaptive confidence interval method to solve this problem. The key idea is to set the threshold according to minimum internal fluctuations.

Given a pair of non-isomorphic graphs $\mathcal{G}$ and $\mathcal{H}$ to be tested. For simplicity, we rename $\mathcal{G}$ as $\mathcal{G}_1$, $\mathcal{H}$ as $\mathcal{G}_2$. For each graph ($\mathcal{G}_1$ and $\mathcal{G}_2$), we generate $p$ groups of graphs, with each group containing $2q$ graphs, represented by:

$$\mathcal{G}_{i,j,k}, \ i \in [2], \ j \in [p], \ k \in [2q]. \tag{10}$$

Similarly, we can calculate $T^2$-statistics for each group ($2p$ groups in total):

$$T^2_{i,j} = q\overline{\boldsymbol{d}}^T_{i,j} \boldsymbol{S}_{i,j} \overline{\boldsymbol{d}}_{i,j}, \ i \in [2], \ j \in [p]. \tag{11}$$

where

$$\overline{\boldsymbol{d}}_{i,j} = \frac{1}{q} \sum_{k=1}^{q} \boldsymbol{d}_{i,j,k}, \ \boldsymbol{d}_{i,j,k} = f(\mathcal{G}_{i,j,k}) - f(\mathcal{G}_{i,j,k+q}), \ i \in [2], \ j \in [p], \ k \in [q],$$

$$\boldsymbol{S}_{i,j} = \frac{1}{q-1} \sum_{j=1}^{q} (\boldsymbol{d}_{i,j,k} - \overline{\boldsymbol{d}}_{i,j})(\boldsymbol{d}_{i,j,k} - \overline{\boldsymbol{d}}_{i,j})^T. \tag{12}$$

Similar to major procedure, we can conduct an $\alpha$-level test of $H_0 : \delta = \boldsymbol{0}$ versus $H_1 : \delta \neq \boldsymbol{0}$, it should always accept $H_0$(the GNN cannot distinguish them) since the $2q$ graphs in each group are essentially the same. And $T^2$-statistics should satisfy the:

$$T^2_{i,j} = q\overline{\boldsymbol{d}}^T_{i,j} \boldsymbol{S}_{i,j} \overline{\boldsymbol{d}}_{i,j} < \frac{(q-1)n}{(q-n)} F_{n,q-n}(\alpha). \tag{13}$$

Table 3: A general theoretical expressiveness upper bound of subgraph with radius $k$

| Radius | 1 | 2 | 3 | 4 | 5 | 6 | 7 | 8 | 9 | 10 |
|---|---|---|---|---|---|---|---|---|---|---|
| #Accurate on BREC | 252 | 298 | 300 | 327 | 326 | 385 | 398 | 398 | 399 | 400 |

If the GNN can distinguish the pair, $T_{\text{test}}^2$ in major procedure and $T_{i,j}^2$ in adaptive confidence interval should satisfy the:

$$T_{\text{test}}^2 > \frac{(q-1)n}{(q-n)} F_{n,q-n}(\alpha) > T_{i,j}^2, \forall i \in [2], \; j \in [p]. \tag{14}$$

Thus we set the adaptive confidence interval as Threshold $= Max_{i \in \{1,2\}, \; p \in \{1,...,P\}} \{T_{i,p}^2\}$. Then we conduct Major Procedure and Reliability Check based on Threshold similar to RPC. The pipeline is shown in Fig 6.

In our analysis of the current evaluation method, we take into account the probabilities of false positives and false negatives. Typically, achieving extremely low levels of both probabilities simultaneously is challenging, and there is often a trade-off between them. However, since false positives can undermine the reliability of the methods, we prioritize establishing stringent bounds for this type of error. On the other hand, false negatives are explained in a more intuitive manner, acknowledging their presence but placing greater emphasis on minimizing false positives.

Regarding false positives, we give the following theorem.

**Theorem E.1.** *The false positive rate with adaptive confidence interval is $\frac{1}{2^{2P}}$.*

*Proof.* We first define false positives more formally. False positives mean the GNN $f$ cannot distinguish $\mathcal{G}$ and $\mathcal{H}$, but we reject $H_0$ and accept $H_1$. $f$ cannot distinguish $\mathcal{G}$ and $\mathcal{H}$ means $f(\mathcal{G}) = f(\mathcal{H}) = f(\mathcal{G}^\pi) \sim \mathcal{N}(\boldsymbol{\mu}_\mathcal{G}, \boldsymbol{\Sigma}_\mathcal{G})$. Since $\boldsymbol{d}_i$ in major procedure and $\boldsymbol{d}_{i,j,k}$ in adaptive confidence interval are derived from paired comparison by same function outputs, i.e., from $f(\mathcal{G})$ and $f(\mathcal{H})$, and from $f(\mathcal{G})$ and $f(\mathcal{G}^\pi)$, respectively. $\boldsymbol{d}_i$ and $\boldsymbol{d}_{i,j,k}$ should follow the same distribution, leading that $T_{\text{test}}^2$ and $T_{i,j}^2$ are independently random variables following the same distribution. Thus $P(T_{\text{test}}^2 > T_{i,j}^2) = \frac{1}{2}$. Then we can calculate the probability of false positives as

$$P(Rejecting \; H_0) = P(T_{\text{test}}^2 > Threshold = Max_{i \in [2], \; j \in [p]} \{T_{i,j}^2\}) = \frac{1}{2^{2p}}. \tag{15}$$

Thus we proof theorem E.1. $\qquad\square$

Regarding false negatives, we propose the following explanation. A small threshold can decrease the false negative rate. Thus without compromising the rest of the theoretical analysis, we give the minimum value of the threshold. Equation 13 introduces a minimum threshold restriction. We obtain the threshold strictly based on it by taking the maximum value, which is the theoretical minimum threshold that minimizes the false negative rate.

# F    Subgraph GNNs

In this section, we discuss settings for subgraph GNN models. The most important setting is the subgraph radius. As discussed before, a larger radius can capture more structural information, increasing the model's expressiveness. However, it will include more invalid information, making reaching the theoretical upper bound harder. Thus we need to find a balance between the two.

To achieve this, we first explore the maximum structural information that can be obtained under a given radius. Following Papp and Wattenhofer [17], we implement $N_k$ method, which embeds the isomorphic type of $k$-hop subgraph when initializing. This method is only available in the theoretical analysis as one can not solve the GI problem by manually giving graph isomorphic type. We mainly use it as a general expressiveness upper bound of subgraph GNNs. The performance of $N_k$ on BREC is shown in Table 3. Actually, $N_3$ already successfully distinguishes all graphs except for CFI graphs. $k = 6$ is an important threshold as $N_k$ outperforms 3-WL (expressiveness upper bound for most subgraph GNNs [15, 16]) in all types of graphs. An interesting discovery is that increasing the radius does not always lead to expressiveness increasing as expected. This is caused by the fact that we only

Table 4: The performance of 3-WL with different iteration times

| Iterations | 1 | 2 | 3 | 4 | 5 |
|---|---|---|---|---|---|
| #Accurate on BREC | 193 | 209 | 217 | 264 | 270 |

Table 5: Substructure-based model performance on BREC

| Model | Basic Graphs (60) | | Regular Graphs (140) | | Extension Graphs (100) | | CFI Graphs (100) | | Total (400) | |
|---|---|---|---|---|---|---|---|---|---|---|
| | Number | Accuracy | Number | Accuracy | Number | Accuracy | Number | Accuracy | Number | Accuracy |
| $S_3$ | 52 | 86.7% | 48 | 34.3% | 5 | 5% | 0 | 0% | 105 | 26.2% |
| $S_4$ | 60 | 100% | 99 | 70.7% | 84 | 84% | 0 | 0% | 243 | 60.8% |
| GSN-v(k=3) | 52 | 86.7% | 48 | 34.3% | 5 | 5% | 0 | 0% | 105 | 26.2% |
| GSN-v(k=4) | 60 | 100% | 99 | 70.7% | 84 | 84% | 0 | 0% | 243 | 60.8% |
| GSN-e(k=3) | 59 | 98.3% | 48 | 34.3% | 52 | 52% | 0 | 0% | 159 | 39.8% |
| GSN | 60 | 100% | 99 | 70.7% | 95 | 95% | 0 | 0% | 254 | 63.5% |

encode the exact $k$-hop subgraph instead of 1 to $k$-hop subgraphs. This phenomenon is similar to subgraph GNNs, revealing the advantages of using distance encoding.

We then test the subgraph GNNs' radii by increasing them until reaching the best performance, which is expected to be a perfect balance. For some methods, radius= 6 is the best selection, which is consistent with the theory. The exceptions are NGNN, NGNN+DE, KPGNN, I$^2$-GNN and SSWL_P. NGNN directly uses an inner GNN to calculate subgraph representation, whose expressiveness is restricted by the inner GNN. As the subgraph radius increases, though the subgraph contains information, the simple inner GNN can hardly give a correct representation. That's why radius= 1 is the best setting for NGNN. NGNN+DE and I$^2$-GNN add distance encodings, making the subgraph with a large radius can always clearly extract a subgraph with a small radius. Therefore, a large radius= 8 is available. KPGNN utilizes a similar setting by incorporating distance to subgraph representation, and radius= 8 is also the best setting. KPGNN can also use graph diffusion to replace the shortest path distance. Though graph diffusion outperforms some graphs, the shortest path distance is generally a better solution. Previous findings reveal the advantages of using distance, which we hope can be more widely used in further research. SSWL_P achieves better expressiveness with theoretical minimum components, making more information available.

# G $k$-WL Hierarchy GNNs

In this section, we discuss settings for $k$-WL hierarchy GNN models. $k$-WL algorithm requires a converged tuple embedding distribution for GI. However, $k$-WL hierarchy GNNs do not have the definition of converging. It will output the final embeddings after a specific number of layers, i.e., the iteration times of $k$-WL. Thus we need to give a suitable number of layers where the $k$-WL converged after the number of iteration times. In theory, increasing the number of layers always leads to a non-decreasing expressiveness, since the converged distribution will not change furthermore. However, more layers may cause over-smoothing, leading to worse performance in practice.

To keep a balance, we utilize similar methods for subgraph GNNs. We first analyze the iteration times of 3-WL, shown in Table 4. One can see 6 iteration times are enough for all types of graphs. Then we increase the layers of $k$-WL GNNs until reaching the best performance. We finally set 5 layers for PPGN, 4 layers for KCSet-GNN and 6 layers for $\delta$-k-LGNN.

# H Substructure-based GNNs

In this section, we discuss the performance of substructure-based GNN models. Specifically, we focus on the GSN (Graph Substructure Network) model proposed by Bouritsas et al. [43], which offers a straightforward neural network implementation, denoted as GSN-v, of the $S_k$ substructure. Additionally, we introduce GSN-e, a slightly stronger version of GSN-v that incorporates features on edges instead of just nodes.

Experimental results presented in Table 5 demonstrate that GSN-v achieves a perfect match with the performance of $S_k$. Furthermore, GSN-e outperforms GSN-v, indicating superior performance when edge features are included.

Table 6: The performance of DropGNN with different sample numbers

| #Samples | 100 | 200 | 400 | 800 | 1200 | 1600 |
|---|---|---|---|---|---|---|
| #Accurate on BREC | 177 | 222 | 242 | 253 | 260 | OOM |

Table 7: Model Hyperparameters

| Model | Radius | Layers | Inner dim | Learning rate | Weight decay | Batch size | Epoch | Early stop threshold |
|---|---|---|---|---|---|---|---|---|
| NGNN | 1 | 6 | 16 | $1e-4$ | $1e-5$ | 32 | 20 | 0.01 |
| DE+NGNN | 8 | 6 | 128 | $1e-4$ | $1e-5$ | 32 | 30 | 0.01 |
| DS-GNN | 6 | 10 | 32 | $1e-4$ | $1e-5$ | 32 | 30 | 0 |
| DSS-GNN | 6 | 9 | 32 | $1e-4$ | $1e-4$ | 32 | 20 | 0.01 |
| SUN | 6 | 9 | 32 | $1e-4$ | $1e-4$ | 32 | 20 | 0.01 |
| SSWL_P | 8 | 8 | 64 | $1e-5$ | $1e-5$ | 8 | 20 | 0.1 |
| GNN-AK | 6 | 4 | 32 | $1e-4$ | $1e-4$ | 32 | 10 | 0.1 |
| KP-GNN | 8 | 8 | 32 | $1e-4$ | $1e-4$ | 32 | 20 | 0.3 |
| I$^2$GNN | 8 | 5 | 32 | $1e-5$ | $1e-4$ | 16 | 20 | 0.2 |
| PPGN | / | 5 | 32 | $1e-4$ | $1e-4$ | 32 | 20 | 0.2 |
| $\delta$-k-LGNN | / | 6 | 16 | $1e-4$ | $1e-4$ | 16 | 20 | 0.2 |
| KC-SetGNN | / | 4 | 64 | $1e-4$ | $1e-4$ | 16 | 15 | 0.3 |
| GSN | / | 4 | 64 | $1e-4$ | $1e-5$ | 16 | 20 | 0.1 |
| DropGNN | / | 10 | 16 | $1e-3$ | $1e-5$ | 16 | 100 | 0 |
| OSAN | / | 8 | 64 | $1e-3$ | $1e-5$ | 16 | 40 | 0 |
| Graphormer | / | 12 | 80 | $2e-5$ | 0 | 16 | 100 | 0 |

# I  Random GNNs

In this section, we delve into the settings for random GNNs. Random GNNs leverage samples from graphs using specific strategies, and both the number of samples and the sampling strategies have an impact on performance.

For DropGNN, the sampling strategy revolves around a relatively straightforward approach of deleting nodes. As for the number of samples, it is recommended to set it to the average number of nodes in the dataset. In our reported results, we set the number of samples to 100, which aligns with the average number of nodes. The ablation study results on the number of samples can be found in Table 6.

Another approach, OSAN, proposes a data-driven method that achieves similar performance with fewer samples. This is achieved by training the model to select diverse samples. However, it requires an additional training framework and may not necessarily lead to improved performance. In our case, we select the edge-deleting strategy and set the number of samples to 20.

# J  Experiment Settings

All experiments were performed on a machine equipped with an Intel Core i9-10980XE CPU, an NVIDIA RTX4090 graphics card, and 256GB of RAM.

**RPC settings.** For non-GNN methods, the output results are uniquely determined, and as such, this part of the experiment does not require RPC. It is worth noting that most non-GNN baselines involve running graph isomorphism testing software on subgraphs, and they mainly serve as theoretical references in our evaluation.

Regarding GNNs, we employ RPC with $q = 32$ and $d = 16$ to evaluate their performance. Considering a confidence level of $\alpha = 0.95$, which is a typical setting in statistics, the threshold should be set to $\frac{(q-1)d}{(q-d)} F_{d,q-d}(\alpha) = 31 F_{16,16}(0.95) = 72.34$.

To ensure robustness, we repeat all evaluation methods ten times using different seeds selected from the set $\{100, 200, \ldots, 1000\}$. We consider the final results reliable only if the model passes the Reliability check for all graphs with any seed, meaning that the quantification of the output embedding distance between isomorphic pairs is always smaller than the threshold. The reported results are selected as the best results rather than the average, as we aim to explore the upper bound of expressiveness.

**Training settings.** We employ a Siamese network design and utilize the cosine similarity loss function. Another commonly used loss function is contrastive loss [34], which directly calculates the difference between two outputs. However, we opt for cosine similarity loss due to its advantage of measuring output difference under the same scale through normalization. This approach prevents model outputs from being excessively amplified, which could otherwise magnify minor precision errors and treat them as differentiated results of the model.

We use the Adam optimizer with a learning rate searched from $\{1e-3, 1e-4, 1e-5\}$, weight decay selected from $\{1e-3, 1e-4, 1e-5\}$, and batch size chosen from $\{8, 16, 32\}$. Graphormer, on the other hand, follows the original training settings on ZINC.

We incorporate an early stopping strategy, which halts training when the loss reaches a small value. While for random GNNs, we do not utilize early stopping. The maximum number of epochs is typically set to around 20 since the model can often distinguish a pair relatively quickly.

**Model hyperparameters.** The most crucial hyperparameters related to expressiveness, such as the subgraph radius for subgraph GNNs and the number of layers for $k$-WL hierarchy GNNs, are determined through theoretical analysis, as outlined in Appendix F and G. These hyperparameters have a direct impact on the expressiveness of the models.

Other hyperparameters also implicitly influence expressiveness. We generally adopt the same settings as previous expressiveness datasets, with two exceptions: inner embedding dimension and batch normalization.

The inner embedding dimension reflects the model's capacity. For smaller and simpler expressiveness datasets used in the past, a small embedding dimension has been sufficient. However, the appropriate embedding dimension for BREC is unknown, so we generally conduct a search within the range of $16, 32, 64, 128$.

Additionally, we utilize batch normalization for all models, even though it may not have been used in all previous models. Batch normalization helps control the outputs within a suitable range, which can be beneficial for distinguishing graph pairs.

The detailed hyperparameter settings for each method are provided in Table 7.

# K   Graph Generation

In this section, we provide an overview of how the graphs in the BREC dataset were generated.

**Basic graphs.** This category consists of 60 pairs of graphs, each containing 10 nodes. To generate these graphs, the 1-WL algorithm was applied to all 11.7 million graphs with 10 nodes, resulting in a hash value for each graph. Among these graphs, 83,074 happened to have identical hash values as others. From this set, 60 pairs of graphs were randomly selected.

**Regular graphs.** This category includes 140 pairs of regular graphs. For the 50 simple regular graphs, the search was conducted for regular graphs with 6 to 10 nodes, and 50 pairs of regular graphs with the same parameters were randomly selected. For the 50 strongly regular graphs, the number of nodes ranged from 16 to 35. The graphs were obtained from sources such as http://www.maths.gla.ac.uk/ es/srgraphs.php and http://users.cecs.anu.edu.au/ bdm/data/graphs.html. For the 20 4-vertex condition graphs, a search was conducted on http://math.ihringer.org/srgs.php, and the simplest 20 pairs of 4-vertex condition graphs with the same parameters were selected. For the 20 distance regular graphs, a search was performed on https://www.distanceregular.org/, and the simplest 20 pairs of distance regular graphs with the same parameters were chosen.

**Extension graphs.** This category consists of 100 pairs of graphs based on comparing results between GNN extensions. The $S_3$, $S_4$, and $N_1$ algorithms were applied to all 1-WL-indistinguishable graphs with 10 nodes. This yielded 4,612 $S_3$-indistinguishable graphs, 1,132 $N_1$-indistinguishable graphs, and 136 $S_4$-indistinguishable graphs. From these sets, 60 pairs of $S_3$-indistinguishable graphs, 20 pairs of $N_1$-indistinguishable graphs, and 10 pairs of $S_4$-indistinguishable graphs were randomly selected. Care was taken to ensure that no graphs were repeated. Additionally, 10 pairs of graphs were added using a virtual node strategy, including 5 pairs obtained by adding a virtual node to a 10-node regular graph and 5 pairs based on $C_{2l}$ and $C_{l,l}$ as described in Papp and Wattenhofer [17].

**CFI graphs.** This category consists of 100 pairs of graphs generated based on the CFI methods proposed by Cai et al. [28]. All CFI graphs with backbones ranging from 3 to 7-node graphs were generated. From this set, 60 pairs of 1-WL-indistinguishable graphs, 20 pairs of 3-WL-indistinguishable graphs, and 20 pairs of 4-WL-indistinguishable graphs were randomly selected.

These different categories of graphs provide a diverse range of graph structures and properties for evaluating the expressiveness of GNN models.