# OpenReview forum: "Towards Better Evaluation of GNN Expressiveness with BREC Dataset"
_NeurIPS.cc/2023/Track/Datasets_and_Benchmarks — Submitted to NeurIPS 2023 Datasets and Benchmarks_

### Official Review · Reviewer_AZWH · 2023-07-01

**Rating:** 6
**Confidence:** 3
**Correctness:** Yes.
**Clarity:** Yes.

**Strengths:**

1. The new datasets are more diverse from different categories with 4-WL-indistinguishable, undistinguishable from the existing most expressive GNN.

2. The new evaluation is more aligned with the theoritical results.

3. Comprehensive experiments indicate the importance of the new proposed datasets and evaluation methods

**Additional Feedback:**

1. I would suggest having more discussion on the opportunity and future direction in the graph expressive domain.
2. What practical value does the graph expressiveness problem have? what new application can this domain inspired.

**Documentation:**

Yes.

**Ethics:**

No.

**Limitations:**

Yes.

**Opportunities For Improvement:**

The paper seems to target the graph expressiveness problem which is an important problem in the theory of GNN. Nonetheless, there is no clear discussion on the practical value of graph expressiveness. As far as I know, previous important literature like "how powerful are graph neural networks" are proposed to improve the GNN expressiveness for better performance in the real-world scenario. Nonetheless, the proposed datasets seem to have no node feature and datasets show no correlation with practical values. It is unclear to me why we require more expressive GNN without considering the generalization ability and practical value. What is the practical benefit of distinguishing between different graphs.

**Relation To Prior Work:**

Yes, previous datasets are discussed carefully.

**Summary And Contributions:**

This paper provides new datasets and evaluation methods for the graph expressiveness problem.  The new datasets and expressive analysis enable experiment verification that the evaluation procedure aligns well with known graph expressive theoretical results.  A reliable empirical comparison serves as a good benchmark for expressiveness analysis.

---

> ### Author Response · Authors · 2023-08-21
> **Response to Reviewer AZWH [1/2]**
>
> >OFI1: “The paper seems to target the graph expressiveness problem which is an important problem in the theory of GNN. Nonetheless, there is no clear discussion on the practical value of graph expressiveness. As far as I know, previous important literature like "how powerful are graph neural networks" are proposed to improve the GNN expressiveness for better performance in the real-world scenario. Nonetheless, the proposed datasets seem to have no node feature and datasets show no correlation with practical values. It is unclear to me why we require more expressive GNN without considering the generalization ability and practical value. What is the practical benefit of distinguishing between different graphs.”
>
> R1: We will reply in 3 aspects: 1. How does expressiveness aid real-world scenarios? 2. Why do we need a synthetic expressiveness dataset to test performance? 3. What can BREC additionally provide to theoretical analysis?
>
> 1. We will show its advantages in examples and real datasets regarding expressiveness. In chemical areas, the structure is essential for property prediction. However, there exist some structures that standard MPNNs can hardly recognize. For example, the benzene ring (6-cycle) is a common and essential molecule component that most simple GNNs cannot recognize. It is inevitable those GNNs might learn some statistical patterns correlated to benzene rings, but they fundamentally fail to understand the structure and fail to make predictions causally. In real-world datasets, expressive GNNs also perform remarkably in molecular datasets. ZINC [1] is a widely used molecular dataset with regression tasks. Currently, the 2 SOTA models are GRIT [2] and N$^2$-GNN [3], both focusing on expressiveness.
> 2. Then we will discuss why we need a new synthetic expressiveness dataset. Previously proposed datasets, EXP, CSL and SR25, are all synthetic and are used widely to test expressiveness. There is also one dataset partly collected from the real world [4, 5, 6], whose task is to perform node-level regression on the number of cycles, cliques, and in total nine types of substructures. However, these results arise with some mistakes. Some methods, like ID-GNN and NGNN, are proven to be unable to count 4-cycles but happen to give good performance (low enough MAE to be considered able to count 4-cycles). It is assumed that 4-cycles happen to correlate strongly with some other kinds of features that can be captured by these GNNs. The wrongly measured 4-cycle distinguishing ability leads to an unreliable expressiveness measurement. That's why we delicately design such high-expressiveness-demanding graphs synthetically, as these graphs tend to have less statistical correlations than real-world graphs.
> 3. Lastly, we show BREC's practical benefits. The previous analysis mainly focused on 'theoretical expressiveness,' which represents the theoretical approximation of the model with strong assumptions (deep enough layers, universal approximation of MLP, etc.). However, whether these models' practical implementations can achieve their theoretical power is constantly questioned and worth studying. Previous expressiveness datasets only served as a simple and incomplete verification, but BREC is designed to test whether the theoretically powerful models can reach their expressiveness in practice. Regarding generalization, it can be measured by real-world performance. And we can compare methods with different expressivity and real-world performance to understand the interplay. Generally, higher expressive power leads to better real-world performance. For example, numerous subgraph-based models show remarkable performance with better expressivity (expressivity results in Section 5 subgraph-based models or [7]; real-world performance see [3] Section 6 Experiments). In some situations, better expressiveness does not lead to better performance. Our dataset can also help us understand this. For example, PPGN is theoretically the same as 3-WL but does not give a promising real-world performance. Our tests firstly show that it cannot reach its theoretical expressive power, which can help explain real-world performance.

---

> > ### Author Response · Authors · 2023-08-21
> > **Response to Reviewer AZWH [2/2]**
> >
> > >AF1: “I would suggest having more discussion on the opportunity and future direction in the graph expressive domain.”
> >
> > R1: A broadly concerning question is the relationship between expressiveness and generalization. Our dataset can help us understand it through further analysis of expressiveness-related performance. Some other future directions include providing more reliable and explainable results for molecules, implementing and utilizing graph algorithm approximations for better reasoning ability, etc. We have revised the paper and discussed them in the introduction and future work.
> >
> > >AF2: “What practical value does the graph expressiveness problem have? what new application can this domain inspired.”
> >
> > R2: For practical value, please refer to OFI1.
> >
> > References:
> >
> > [1] Dwivedi et al. Benchmarking Graph Neural Networks https://arxiv.org/abs/2003.00982
> >
> > [2] Ma et al. Graph Inductive Biases in Transformers without Message Passing http://arxiv.org/abs/2305.17589
> >
> > [3] Feng et al. Towards Arbitrarily Expressive GNNs in O(n2) Space by Rethinking Folklore Weisfeiler-Lehman https://arxiv.org/abs/2306.03266
> >
> > [4] Huang et al. Boosting the Cycle Counting Power of Graph Neural Networks with I$^2$-GNNs http://arxiv.org/abs/2210.13978
> >
> > [5] Zhao et al. From Stars to Subgraphs: Uplifting Any GNN with Local Structure Awareness http://arxiv.org/abs/2110.03753
> >
> > [6] Chen et al. Can Graph Neural Networks Count Substructures? http://arxiv.org/abs/2002.04025
> >
> > [7] Zhang et al. A Complete Expressiveness Hierarchy for Subgraph GNNs via Subgraph Weisfeiler-Lehman Tests http://arxiv.org/abs/2302.07090

---

### Official Review · Reviewer_vLAp · 2023-07-14
**In general a good paper: proposing the BREC, a dataset for evaluating the expressiveness of GNNs**

**Rating:** 8
**Confidence:** 3
**Correctness:** Yes
**Clarity:** Yes

**Strengths:**

The paper is concise and clear, the BREC dataset proposed is a good mix of different categories of graphs. It's very nice to see that the authors evaluated on large number of models (23 of them).

**Additional Feedback:**

No additional feedback.

**Documentation:**

Yes

**Limitations:**

It's nice to have 4 different categories of graphs in BREC: Basic Graphs, Regular Graphs, Extension Graphs and CFI Graphs. But it would be good to have some examples about how these graphs are related to the graphs that the researchers use for training, like: molecular graphs, protein graphs, social networks graphs etc. and what does good performance on a certain type of graphs mean in real graphs used for training.

**Opportunities For Improvement:**

1, for Extension Graphs and CFI Graphs, the authors mentioned that these were inspired from previous authors (Papp and Wattenhofer and Cai et al.), would be good to mention what's the difference/novelty these datasets compared to previous ones.
2, on line 270: would be useful to add more clarification here about what does reindexing mean. I think it means reindexing the nodes of the graphs?
3, Would be helpful to the reader to understand the theory behind equations 4 to 7 by adding more explanation/details in the appendix.
4, Foe model selection, would be good to see some results on hybrid models like GPS and GPS++.

**Relation To Prior Work:**

Yes

**Summary And Contributions:**

The authors:
1, proposed a dataset called BREC for GNN expressiveness valuation.
2, They also used a new evaluation method called RPC.
3, 23 models were evaluated on BREC using RPC, showing their expressiveness on different sub-categories of BREC and overall evaluation score of BREC.

---

> ### Author Response · Authors · 2023-08-21
> **Response to Reviewer vLAp**
>
> >OFI1: “for Extension Graphs and CFI Graphs, the authors mentioned that these were inspired from previous authors (Papp and Wattenhofer and Cai et al.), would be good to mention what's the difference/novelty these datasets compared to previous ones.”
>
> R1: Thanks for your suggestion. A key difference is that the stage where k-WL fails to distinguish differs. For widely used regular graphs where each node has the same number of neighbors, WL tests converge since initialization, i.e., we can not observe changes through the iterative process of WL. However, Extension Graphs and CFI Graphs are less symmetric for us to observe the process and discover hidden properties. Specifically, the Extension graphs utilize the expressiveness comparison results between several GNN extensions (theoretical approximations of some GNNs) towards a more fine-grained measurement between 1-WL and 3-WL. CFI graphs are constructed with a 'backbone' and 'components' with arbitrary difficulties.
>
> >OF2: “on line 270: would be useful to add more clarification here about what does reindexing mean. I think it means reindexing the nodes of the graphs?”
>
> R2: Thanks for your suggestion. Reindexing means reindexing the nodes of the graphs. We have added a clarification.
>
> >OF3: “Would be helpful to the reader to understand the theory behind equations 4 to 7 by adding more explanation/details in the appendix.”
>
> R3: Thanks for your suggestion. This part mainly arises from utilizing Hotelling's T-square test in Paired Comparisons which has complete analysis and proof (e.g., see [https://en.wikipedia.org/wiki/Hotelling's_T-squared_distribution](https://en.wikipedia.org/wiki/Hotelling%27s_T-squared_distribution) ). Thus we do not introduce detailed theorems. The core theory is that $T^2$ satisfies F-distribution under Assumption 4.1. Now we further add explanations and references for understanding.
>
> >OF4: “Foe model selection, would be good to see some results on hybrid models like GPS and GPS++.”
>
> R4: Thanks for your suggestion. Each layer in GPS includes two parts, an MPNN with both node and edge features and a global attention layer with only node features. Therefore, the global attention layer does not contribute to expressiveness without edge connection information. The expressiveness of GPS equals to the MPNN with encodings. The original experiment also verifies that. (Ablation study on ZINC shows that removing the attention layer has little effect on performance: 0.070→0.071. While removing the MPNN decreases performance sharply: 0.070→0.217). Following ZINC settings, we use RWSE encoding and test performance of GPS and its theoretical approximation (RWSE-WL).
>
> |         | Basic Graph | Regular Graph | Extension Graph | CFI Graph | All     |
> | ------- | ----------- | ------------- | --------------- | --------- | ------- |
> | RWSE-WL | 60/60       | 50/140        | 100/100         | 9/100     | 219/400 |
> | GPS     | 60/60       | 50/140        | 100/100         | 7/100     | 217/400 |
>
> >Limitations 1: “It's nice to have 4 different categories of graphs in BREC: Basic Graphs, Regular Graphs, Extension Graphs and CFI Graphs. But it would be good to have some examples about how these graphs are related to the graphs that the researchers use for training, like: molecular graphs, protein graphs, social networks graphs etc. and what does good performance on a certain type of graphs mean in real graphs used for training.”
>
> R1: Thanks for your question. Generally, graphs in chemical areas tend to have high expressiveness demand. An example is the benzene ring (6-cycle), a common and essential molecule component that most simple GNNs cannot recognize. Performance on datasets also reflects this. ZINC [1] is a widely used molecular dataset with regression tasks, and the current 2 SOTA models, GRIT [2] and N2-GNN [3], both focused on expressiveness. Regarding certain types of graphs in BREC, it is assumed that Basic graphs and Extension graphs tend to reflect performance on relatively small graphs, and regular graphs can reflect situations where graphs are dense and symmetric, while CFI graphs may reflect performance on a large scale and long-range dependency graphs.
>
> References:
>
> [1] Dwivedi et al. Benchmarking Graph Neural Networks https://arxiv.org/abs/2003.00982
>
> [2] Ma et al. Graph Inductive Biases in Transformers without Message Passing http://arxiv.org/abs/2305.17589
>
> [3] Feng et al. Towards Arbitrarily Expressive GNNs in O(n2) Space by Rethinking Folklore Weisfeiler-Lehman https://arxiv.org/abs/2306.03266

---

> > ### Comment · Reviewer_vLAp · 2023-08-26
> > **Thanks**
> >
> > Thanks for the responses and clarification. I'm happy to keep my current rating: 8: Top 50% of accepted papers, clear accept.

---

> > > ### Author Response · Authors · 2023-08-27
> > > **Appreciate Your Affirmation and Suggestions**
> > >
> > > Thank you for giving very positive and encouraging feedback. We truly appreciate your effort in helping us to strengthen the paper and your support for our work!

---

### Official Review · Reviewer_kpAj · 2023-07-19
**Documentation and code can be improved**

**Rating:** 7
**Confidence:** 5

**Strengths:**

The main strength of the paper lies in presenting a new data set that
clearly fills a gap in the literature. Expressivity analyses in most,
if not all, graph learning papers is substantially limited, since the
selection of graphs used to assess expressivity is not diverse. Thus,
this paper makes a highly relevant contribution, viz. suggesting sets
of graphs of varying complexities.

**Additional Feedback:**

I have some minor points that could be addressed in a revision:

- 'be NP' --> 'be an NP-intermediate problem.'

- 'Then a more challenging' --> 'A more challenging'

- 'A methodology' --> 'A method'

- The cosine similarity is technically not a metric in the mathematical
  sense. I would thus refrain from using terms like 'distance.'

- 'If exists' --> 'If there exists'

**Clarity:**

- Next to some language issues, the main point of clarity concerns the
  reporting of the data set size. While I understand that the paper
  wants to clarify that BREC contains more (and more relevant) graphs
  than existing data sets, I believe that switching between 'total
  number of graphs' and 'pairs of graphs' can be confusing at times. As
  a simple example: suppose I only have 30 distinct or unique graphs in
  my data set, it would be confusing to readers to report that the data
  set consists of 435 graphs instead---even though it would of course be
  correct to state that the data set can form 435 pairs. I would
  strongly suggest to go again carefully over the paper to reassess and
  recheck statements pertaining to the data set size.

- Moreover, the construction of the data set, especially the different
  graph types, could be extended. Upon my first reading of the paper,
  I stumbled over the 'CFI' abbreviation; it became clear when reading
  the bibliography. Please spell out this acronym briefly when
  introducing the graphs for the first time.

**Correctness:**

The claims in the paper are correct to my understanding. I have some
concerns about the utility of the evaluation metric, but I raised them
above already.

**Documentation:**

Documentation of the data set is adequate, but as outlined above, an
integration into standard frameworks would improve adoption. Moreover,
the missing versioning will make it hard to track any potential future
updates to the data set, thus precluding additional community
contributions.

**Ethics:**

The data set does not suffer from any ethics concerns.

**Limitations:**

The main limitation of the paper lies in providing data sets that are
*only* exhibiting structural information. This is a limitation shared by
all data sets that deal with expressivity analyses, but it would be
worth mentioning in a separate section of the paper.

**Opportunities For Improvement:**

The current write-up suffers from some issues that need to be rectified
for the paper to be ready for publication:

1. Language: the paper would benefit from additional 'word-smithing,'
   and additional revisions. In the present version, the use of
   non-standard grammar decreases the accessibility and, most
   importantly, the readability of the paper.

2. Code: the code accompanying the paper would benefit from a better
   integration into standard APIs. The examples provided in the
   repository are reasonably easy to follow and try out, but an
   integration of the data set into a framework such as
   `pytorch-geometric` would really help to improve the adoption of the
   data set. In particular, providing pre-defined data loaders for
   classification tasks would be very helpful. This would also be
   a great opportunity for providing a reference implementation of new
   reference metric.

3. Versioning: the data set has to be downloaded from a GitHub
   repository as a `.zip` file. Together with the aforementioned missing
   code integration into standard frameworks, this prevents future
   updates to the data set, such as the integration of additional graphs
   or the correction of any errors.

4. Evaluation: the new evaluation metric shown in Equation 1 should be
   analysed in a more theoretic fashion. Given no *a priori* knowledge
   about models, I wonder to what extent the use of an inner product
   between representations is justified. It would improve the paper if
   this aspect was commented on. As an example of alternative measures,
   it might be interesting to contrast the proposed evaluation method to
   [centred kernel alignment](https://arxiv.org/pdf/1905.00414.pdf), for
   instance.

**Relation To Prior Work:**

Related work is cited adequately and framed adequately, to the best of
my knowledge. However, the discussion of strongly-regular graphs could
be extended; there are more data sets available than the ones cited in
the paper. See the [collection by Brendan McKay](https://users.cecs.anu.edu.au/~bdm/data/graphs.html), for instance.

**Summary And Contributions:**

This paper introduces a new data set for analysing the expressivity of
graph neural networks. Going beyond existing work, this paper presents
a curated set of graphs with *known* hardness properties, ranging from
graphs that can be distinguished by 1-WL to graphs that require 4-WL.

---

> ### Author Response · Authors · 2023-08-21
> **Response to Reviewer kpAj [1/2]**
>
> >OFI1: “Language: the paper would benefit from additional 'word-smithing,' and additional revisions. In the present version, the use of non-standard grammar decreases the accessibility and, most importantly, the readability of the paper.”
>
> R1: Thanks for your suggestions. We have corrected the mistakes and further checked possible presentation issues in the updated paper.
>
> >OFI2: “Code: the code accompanying the paper would benefit from a better integration into standard APIs. The examples provided in the repository are reasonably easy to follow and try out, but an integration of the data set into a framework such as `pytorch-geometric` would really help to improve the adoption of the data set. In particular, providing pre-defined data loaders for classification tasks would be very helpful. This would also be a great opportunity for providing a reference implementation of new reference metric.”
>
> R2: Thanks for your suggestions. We have distributed BREC to Pypi https://pypi.org/project/brec/ for easier usage. And now, it can be easily tested with users' pipelines in a pytorch-geometric manner. We are also looking forward to better integration into standard libraries.
>
> >OFI3: “Versioning: the data set has to be downloaded from a GitHub repository as a `.zip` file. Together with the aforementioned missing code integration into standard frameworks, this prevents future updates to the data set, such as the integration of additional graphs or the correction of any errors.”
>
> R3: We released the dataset customized construction pipeline at https://github.com/GraphPKU/BREC#customize-brec-dataset
> . For the integration of additional graphs or correction of errors, they can be easily implemented by adding original graphs. Further organization and transformation to pyg format will be automated. We will keep upgrading the files and the Pypi package.
>
> >OFI4: “Evaluation: the new evaluation metric shown in Equation 1 should be analysed in a more theoretic fashion. Given no *a priori* knowledge about models, I wonder to what extent the use of an inner product between representations is justified. It would improve the paper if this aspect was commented on. As an example of alternative measures, it might be interesting to contrast the proposed evaluation method to [centred kernel alignment](https://arxiv.org/pdf/1905.00414.pdf), for instance.”
>
> R4: Thanks for your suggestions. In our evaluation process, we use two metrics. The first (Equation 1) arises from contrastive learning settings aiming to enlarge the distance between two non-isomorphic graphs. Compared to contrastive learning regarding alignment and uniformity, our training only focuses on one pair (no augmented views) and does not require such property. The second one (Equation 2-9) arises from paired comparison aiming to control the fluctuation errors.  By surveying articles related to compared learning and similarity evaluation metrics, etc., we found that their settings and goals differ from ours. Therefore, analysis in this area has yet to be included at this time. We will focus on that aspect as well.

---

> > ### Author Response · Authors · 2023-08-21
> > **Response to Reviewer kpAj [2/2]**
> >
> > >Limitations1: “The main limitation of the paper lies in providing data sets that are *only* exhibiting structural information. This is a limitation shared by all data sets that deal with expressivity analyses, but it would be worth mentioning in a separate section of the paper.”
> >
> > R1: Thanks for your suggestion. We already discuss feature usages in Appendix B. Generally, we adhere to model settings and insert corresponding features when required. However, we do not add additional features since they reduce the difficulty of graphs. We will also consider a better scheme to add additional features.
> >
> > >Clarity1: “Next to some language issues, the main point of clarity concerns the reporting of the data set size. While I understand that the paper wants to clarify that BREC contains more (and more relevant) graphs than existing data sets, I believe that switching between 'total number of graphs' and 'pairs of graphs' can be confusing at times. As a simple example: suppose I only have 30 distinct or unique graphs in my data set, it would be confusing to readers to report that the data set consists of 435 graphs instead---even though it would of course be correct to state that the data set can form 435 pairs. I would strongly suggest to go again carefully over the paper to reassess and recheck statements pertaining to the data set size.”
> >
> > R1: Thanks for your suggestion. We use a total of 800 non-isomorphic graphs organized pair-wise to construct 400 pairs (each graph appears only once). Thus, generating 319,600 pairs of graphs (from CSL and SR25 settings) or even more by adding noisy components (from EXP settings) will be easy. However, we limit the number to 400 pairs (each graph appears only once) to keep the most essential and difficult comparisons. And we have revised related expressions.
> >
> > >Clarity2: “Moreover, the construction of the data set, especially the different graph types, could be extended. Upon my first reading of the paper, I stumbled over the 'CFI' abbreviation; it became clear when reading the bibliography. Please spell out this acronym briefly when introducing the graphs for the first time.”
> >
> > R2: Thanks for your suggestion. We already give a more detailed discussion on graph generation in Appendix K and add a link to it when first discussing graph construction. We also add a footnote to describe CFI briefly.
> >
> > >Relation To Prior Work1: “Related work is cited adequately and framed adequately, to the best of my knowledge. However, the discussion of strongly-regular graphs could be extended; there are more data sets available than the ones cited in the paper. See the [collection by Brendan McKay](https://users.cecs.anu.edu.au/~bdm/data/graphs.html), for instance.”
> >
> > R1: Thanks. We use this collection and mention it in Appendix K. We will also consider arranging the main content and appendix to clarify details.
> >
> > >Additional Feedback1: “I have some minor points that could be addressed in a revision:…”
> >
> > R1: We have already revised them and updated ''distance'' to ''difference'’

---

> > ### Comment · Reviewer_kpAj · 2023-08-25
> >
> > Thank you very much for a comprehensive rebuttal and the many improvements! One additional clarification from my side: how is the version of the package tied to the data set? Is the idea that, say, for a major release (i.e. going from `v.1.a.b` to `v.2.x.y`), the data set itself is allowed to change? My point about the version is that, supposing two papers use 'the' BREC data set—one from 2023, the other forthcoming form 2024. If there are any changes to the dataset between the two publication dates, how can we mark this in the paper?

---

> > > ### Author Response · Authors · 2023-08-26
> > > **Response to Possible Modification and Version Problem**
> > >
> > > > Thank you very much for a comprehensive rebuttal and the many improvements! One additional clarification from my side: how is the version of the package tied to the data set? Is the idea that, say, for a major release (i.e. going from v.1.a.b to v.2.x.y), the data set itself is allowed to change? My point about the version is that, supposing two papers use 'the' BREC data set—one from 2023, the other forthcoming form 2024. If there are any changes to the dataset between the two publication dates, how can we mark this in the paper?
> > >
> > > Thanks for your comment. Generally, we will keep the dataset unchanged in a major version. Regarding changes in datasets, firstly, we can split changes to datasets into two types. The first type is 'modification', including regenerating labels, changing features, correcting splits, etc. Regarding this type, the previous comparison results will no longer apply to the new ones, and a new dataset will be published. For example, revised MD17 (https://figshare.com/articles/dataset/Revised_MD17_dataset_rMD17_/12672038) is generated based on MD17 by cleaning label noises; PCQM4Mv2 (https://ogb.stanford.edu/docs/lsc/pcqm4mv2/) is based on PCQM4Mv2 by modifying features and removing some data points. And researchers will consider they are different datasets and make comparisons only under the same one. If modifications of this type are applied to BREC, we will also distribute and update the testing results. The second type is 'addition', like adding new data points. This will also affect the trained model for previous datasets and require distributing new versions. However, since BREC tests on graph pairs independently, we can add new graph pairs, which will not affect the previous test results. If modifications of this type are applied to BREC, we will generate new results for the added graphs only and keep the remaining ones unchanged. Finally the aggregated results will be calculated.

---

> > > > ### Comment · Reviewer_kpAj · 2023-08-26
> > > >
> > > > To clarify my question: what type of versioning scheme do you intend to follow with BREC? How will the 'official' BREC data set by versioned and how will changes be reported?

---

> > > > > ### Author Response · Authors · 2023-08-26
> > > > > **Semantic Versioning Applied to BREC**
> > > > >
> > > > > >To clarify my question: what type of versioning scheme do you intend to follow with BREC? How will the 'official' BREC data set by versioned and how will changes be reported?
> > > > >
> > > > > Thanks for your comment. We will manage BREC adhering to Semantic Versioning, which ensures that any changes to the dataset will lead to a major version update. And the Pypi package will be updated synchronously. The CHANGELOG is available at https://github.com/GraphPKU/BREC/blob/Release/CHANGELOG.md.

---

> > > > > > ### Comment · Reviewer_kpAj · 2023-08-27
> > > > > >
> > > > > > Thank you! This addresses all my comments, and I will adjust my rating accordingly. Good luck with the submission!

---

> > > > > > > ### Author Response · Authors · 2023-08-27
> > > > > > > **Appreciate Your Affirmation and Suggestions**
> > > > > > >
> > > > > > > Thank you for giving very positive and encouraging feedback. We truly appreciate your effort in helping us to strengthen the paper and your support for our work!

---

### Official Review · Reviewer_iUXu · 2023-07-21
**An Interesting Evaluation Idea for GNNs and a Corresponding Dataset**

**Rating:** 8
**Confidence:** 4

**Strengths:**

The authors argue convincingly that classification tasks are problematic to evaluate expressiveness of GNNs, as they either restrict benchmarks to only a few nonisomorphic target graphs or introduce adverse effects due to (massive-) multiclass prediction.
Hence they introduce a contrastive task where the goal is to learn distinct representations for non-isomorphic graphs.
This removes the dependency on additional MLP parts after the graph pooling step.

The dataset is carefully constructed to contain diverse pairs of graphs which are known to be impossible to distinguish for some GNN architectures. I expect the dataset and the proposed evaluation protocol to help the community assessing its progress.

**Additional Feedback:**

Generally, I would like to know the actual number of nonisomorphic graphs in your dataset to be reported somewhere prominently in your work. It might be somewhere between 29 and 800, which makes a difference wrt. diversity of the dataset.

Minor Comments:
- l 3: what do you mean by 'uniform'?
- l 53: 'discimination difficulty'? I don't understand this word. Furthermore, it is not really clear that some generator must always produce pairs of graphs that some given model can completely distinguish or cannot. Think of a pair that is not 2-WL distinguishable. It may be 3-WL distinguishable, or it may not. By accident, you may create harder exaples.
- l 54: 'the performance of GNN variants falls either at 0 (completely indistinguishable).' How do you measure performance, here. Typically, accuracy will be higher than 0 even in this case.
- references: there are several references giving only arxiv links, while the papers are published. E.g. Xu et al. 2019

**Clarity:**

The paper is well-written and mostly easy to follow.
After reading the main part of the paper, only the training details remain somewhat elusive.
It would be beneficial to specify the required experimental setup in more detail to allow comparison of results.

**Correctness:**

The dataset is constructed in a well-structured way. It presents pairs of graphs that were constructed in multiple different ways to offer difficult instances for a broad range of graph learning methods.

**Documentation:**

From Skimming through the documentation on github, I don't feel too confident to be able to run my own model within your evaluation in less than a day.
I think the documentation should be improved to allow an easy access to the repository.
Helpful for me would be:
- Setup information:
	- Which packages are required, or is there an `environment.yaml` for conda or similar?
	- what do I need to do to reproduce your experiments?
- A simple howto to get started with my own model. Maybe it suffices to say, 'copy this folder and replace the model in line X in that file with yours'?
- A pointer to a data loader for pytorch geometric, if I just want to get the data, not your evaluation setup.

**Ethics:**

No.

**Limitations:**

The authors include statistical tests to increase the reliability of their evaluation protocol which is based on comparing (noisy) vectors of floating point numbers.

**Opportunities For Improvement:**

While the dataset is much larger than other synthetic datasets in this domain, 400 pairs of graphs are not very large. (I can get 406 pairs by looking at all possible pairs among 29 graphs).
The focus on reporting only the number of pairs in the dataset is sometimes confusing.
For example, I wonder if training really happens on a subset of the 400 pairs. An alternative to improve the input size would be to combine graphs from different pairs which can likely also be assumed to be non-isomorphic, to improve the overall number of pairs at hand, at least during training?

**Relation To Prior Work:**

The authors discuss several other synthetic datasets for GNN expressivity evaluation and are well-oriented in recently proposed 'powerful' GNN architectures.

**Summary And Contributions:**

The authors introduce the synthetic BREC dataset to evaluate the expressiveness of graph neural networks.
The dataset is organized in 400 pairs of nonisomorphic graphs of various `difficulties` and the task is contrastive representation learning to distinguish the pairs.
Empirical experiments on 23 GNN and non-GNN approaches show that  BREC is suitable to show performance differences among (GNN) methods and hence can serve as a suitable benchmark.

---

> ### Author Response · Authors · 2023-08-21
> **Response to Reviewer iUXu [1/2]**
>
> >OFI1: "While the dataset is much larger than other synthetic datasets in this domain, 400 pairs of graphs are not very large. (I can get 406 pairs by looking at all possible pairs among 29 graphs). The focus on reporting only the number of pairs in the dataset is sometimes confusing. For example, I wonder if training really happens on a subset of the 400 pairs. An alternative to improve the input size would be to combine graphs from different pairs which can likely also be assumed to be non-isomorphic, to improve the overall number of pairs at hand, at least during training?"
>
> R1:
> 1. Regarding sizes, our dataset actually owns 800 non-isomorphic and high-expressiveness-demanding graphs. Thus, generating 319,600 pairs of graphs (from CSL and SR25 settings) or even more by adding noisy components (from EXP settings) is easy. However, we limit the number to 400 pairs (each graph appears only once) to keep the most essential and difficult comparisons. For further explanation, please refer to General Response 2.
> 2. Regarding reporting format, we report the number of pairs because all the tests are individually done on each pair. For each pair, we reset model parameters and train the model with contrastive loss to enlarge the distance between the two graphs' output. Then we utilize a paired-comparison-inspired evaluation method to give a reliable test result to tell whether our model can distinguish the pair. Therefore, we can obtain an accurate number of pairs the model can distinguish. Since we do not adopt the traditional classification setting as in CSL/EXP, we cannot combine graphs from different pairs to improve the training size.
> 3. Although naively generating 319,600 pairs is not interesting (since most pairs are too easy to distinguish), we can still limit the pair generation to certain types of graphs. For example, we can still follow the contrastive learning paradigm while operating on all pairs of Basic graphs, i.e., we train one model that can simultaneously distinguish all Basic graphs. Nevertheless, this will make the evaluation much longer. Thanks for the suggestion. We will consider it in our future work.
>
> >Doc1: “Which packages are required, or is there an environment.yaml for conda or similar?”
>
> R1:
> 1. We provide the required packages with tested combination versions at https://github.com/GraphPKU/BREC#requirements, including numpy, pytorch, torch_geometric, networkx, loguru, etc.
> 2. Currently, we distribute BREC to Pypi https://pypi.org/project/brec/1.0.0/ so that it can also be installed by `pip install brec` if using pypi package.
> 3. We do not explicitly give requirements regarding package versions because we hope models can be seamlessly tested on BREC without package conflicts. During our experiment, the tested models' original packages' versions satisfy BREC, and only several packages need to be additionally installed (already posted on README in corresponding directories). The only exception is SUN, with an old PyTorch version lacking a covariance calculating function (now SUN’s new implementation has already used the new PyTorch version). So we implement it in SUN-related test codes.
>
> >Doc2: “what do I need to do to reproduce your experiments?”
>
> R2: All reproducing steps are available at https://github.com/GraphPKU/BREC#reproduce-baselines. The steps are: 1) Select one model and search its corresponding directory. 2) Install packages following the corresponding README. 3) Run python test_brec_search.py
>
> >Doc3: “A simple how to to get started with my own model. Maybe it suffices to say, 'copy this folder and replace the model in line X in that file with yours'?”
>
> R3: Thanks for your suggestions.
> 1. A simple how-to get started is available at https://github.com/GraphPKU/BREC#test-your-own-gnn. We also provide Pypi distribution at https://pypi.org/project/brec/1.0.0/ with a simple example.
> 2. The steps are: 1) Separate your original pipeline to offline operation, dataset processing and model construction. It is recommended to use `pre_transform` and `transform` to process datasets. 2) If using evaluation files in GitHub, please implement `test_BREC.py` with corresponding functions `pre_calculation`, `get_dataset` and `get_model`. Using Pypi distribution, you can call `brec_dataset` and `evaluator` in your implementation. Once prepared, simply running `test_BREC_search.py` will return the final result.
>
> >Doc4: “A pointer to a data loader for pytorch geometric, if I just want to get the data, not your evaluation setup.”
>
> R4:
> 1. The most direct method to obtain the dataset is using `pip install brec` and `from brec.dataset import BrecDataset`, or using `get_dataset` in `test_BREC.py`
> 2. For better visualization, you can also get raw graphs in graph6 format (widely supported in torch_geometric, networkx, etc.) of each type. A simple guide is at https://github.com/GraphPKU/BREC#customize-brec-dataset.

---

> > ### Author Response · Authors · 2023-08-21
> > **Response to Reviewer iUXu [2/2]**
> >
> > >AF1: “Generally, I would like to know the actual number of nonisomorphic graphs in your dataset to be reported somewhere prominently in your work. It might be somewhere between 29 and 800, which makes a difference wrt. diversity of the dataset.”
> >
> > R1: The actual number is 800, as reported in Table 1. In other words, all graphs in our dataset are distinct. We now highlighted it in the revised version.
> >
> > >AF2: “l 3: what do you mean by 'uniform'?”
> >
> > R2: Here, we want to emphasize the incomplete analysis and incomparability of some GNNs. By uniform, we mean k-WL builds a complete hierarchy, with k as a parameter to adjust expressiveness (expressiveness monotonically increasing with k increasing), and some methods exist to generate a bunch of graphs that fail to be distinguished by k-WL. However, for some expressive GNNs utilizing some tricks, there is no such frameworks that can strictly characterize their expressiveness. For example, one method may partially outperforms 3-WL (or 2-FWL) by showing its advantage in some strongly regular graphs. However, whether it can match 3-WL in other sophisticated situations or how to compare it with other expressive GNNs is questionable and worth studying. In contrast, our dataset can help build an empirical expressiveness hierarchy uniformly suiting various methods.
> >
> > >AF3: “l 53: 'discimination difficulty'? I don't understand this word. Furthermore, it is not really clear that some generator must always produce pairs of graphs that some given model can completely distinguish or cannot. Think of a pair that is not 2-WL distinguishable. It may be 3-WL distinguishable, or it may not. By accident, you may create harder exaples.”
> >
> > R3: By discrimination difficulty, we mean the level of the algorithm to distinguish the graphs. (EXP: all graphs are 1-WL or 2-WL indistinguishable, but 2-FWL or 3-WL distinguishable. CSL: all graphs are 1-WL or 2-WL indistinguishable, but 2-FWL or 3-WL distinguishable. SR25: all graphs are 2-FWL or 3-WL indistinguishable, but 3-FWL or 4-WL distinguishable). Regarding the exact difficulties, we also implement k-WL and verify that the generated graphs match their precise difficulty bound.
> >
> > >AF4: “l 54: 'the performance of GNN variants falls either at 0 (completely indistinguishable).' How do you measure performance, here. Typically, accuracy will be higher than 0 even in this case.”
> >
> > R4: Previous datasets are generally transformed into a classification problem and measured by accuracy. So even random guess can have a nonzero accuracy. Here we use 0 to mean the GNN will output the same embeddings for all graphs, and the accuracy equals random guessing (not saying the accuracy is 0). Sorry for the confusion. We have revised the expression from '0' to 'random guessing'.
> >
> > >AF5: “references: there are several references giving only arxiv links, while the papers are published. E.g. Xu et al. 2019”
> >
> > R5: Sorry for the format problem. We have revised the references.

---

> > > ### Comment · Reviewer_iUXu · 2023-08-30
> > >
> > > Thank you for your detailed replies. They are on point and I am even more happy to recommend acceptance with the changes in place.

---

> > > > ### Author Response · Authors · 2023-08-30
> > > > **Appreciate Your Affirmation and Suggestions**
> > > >
> > > > Thank you for giving very positive and encouraging feedback. We truly appreciate your effort in helping us to strengthen the paper and your support for our work!

---

### Official Review · Reviewer_Zvmi · 2023-07-24

**Rating:** 4
**Confidence:** 3
**Correctness:** I

**Strengths:**

- Having a fixed set of benchmark datasets for investigating the expressive power of GNNs and related architectures from an empirical standpoint is a meaningful contribution to the GNN community. It allows researchers quickly understand if their theoretical expressiveness results translate into practice. Hence the paper somehow reduces the gap between theoretical expressivity from the k-WL lense and practical expressivity (i.e., GNNs trained with SGD).
- The authors put a lot of effort into comparing different GNNs with different levels of expressive power.

**Additional Feedback:**

Not applicable.

**Clarity:**

It is okay, but several parts could be improved, especially section 4 is hard to follow.

**Documentation:**

Seems adequate.

**Ethics:**

Not applicable.

**Limitations:**

Not applicable.

**Opportunities For Improvement:**

- There are some formal mistakes, e.g.,
   - l. 86: You seem to be missing that $\phi$ needs to be a bijection
   - l. 89: "However, GI is suspected to be NP". This does not make sense. GI is clearly in NP. You might mean NP-complete. It is unknown whether GI is in P, in NP-complete, or somewhere in between.
   - l. 90: "iterating all n! permutations to test the bijection", No, if you have given the bijection, you can test in linear time if the bijection is an isomorphism. This also implies that the problem is in NP.
- The dataset is all taken from the literature, i.e., no really new datasets.
- The dataset is rather small (800 graphs). Hence, it is unclear to what extent the dataset is useful for understanding the interplay of expressive power and generalization.
- The authors should make it clear what using the dataset gives a user over just theoretically analyzing the expressive power.
- The need for the theoretical framework is not clear. That is, is it really the case that previous evaluations suffer from numerical precision problems?

**Relation To Prior Work:**

Discussion of related work is adequate. The authors make clear that the datasets are based on existing works and cited when necessary.

**Summary And Contributions:**

The paper proposes a benchmark dataset for evaluating the expressive power of message-passing GNNs and their more powerful extensions from an empirical point of view. The authors propose the BREC dataset consisting of 400 pairs of non-isomorphic graphs from four different difficulty levels (in terms of distinguishing with the k-WL). The graphs are either taken from the literature or are variations of such graphs. Moreover, they propose an evaluation procedure for better understanding differences in model performance.

---

> ### Author Response · Authors · 2023-08-21
> **Response to Reviewer Zvmi [1/2]**
>
> >OFI1: "There are some formal mistakes…"
>
> R1: Thanks for your suggestions. We have corrected the mistakes and further checked possible presentation issues in the updated paper.
>
> >OFI2: "The dataset is all taken from the literature, i.e., no really new datasets."
>
> R2:
>
> 1. We actually give many new graphs. Basic graphs and Extension graphs are generated from exhaustively searching among all possible 10-node graphs with the corresponding expressivity level, which are new for the community. Regular graphs include four subcategories, where 4-vertex condition graphs are used in expressiveness for the first time and distance regular graphs are also large-scalely used in expressiveness for the first time (previous research only used one simple example, which degenerates to strongly regular graphs). CFI graphs are generated by running the CFI algorithm on backbone graphs up to 7 nodes with desired difficulty. We admit that many graphs have been existing and analyzed theoretically before, but to our best knowledge, we are the first to actually generate them in a large scale.
> 2. Unlike datasets from the real world that can be sampled randomly and broadly, our dataset requires all the graphs to be **understood and analyzed** from the expressivity perspective. Therefore, we cannot fully discard the current expressivity frameworks and generate new graphs arbitrarily for testing expressivity (since 1-WL can already distinguish almost all random graphs).
> 3. We implement high-dimensional WL and FWL with any dimension, all the GNN extension algorithms proposed by Papp, and the CFI algorithm by ourselves. To our best knowledge, neither those theoretical papers nor widely-used graph-related software gives such implementations, and we are the first one to provide publicly available open-source implementations. We believe these will be extremely valuable for future expressvity research.
>
> >OFI3: "The dataset is rather small (800 graphs). Hence, it is unclear to what extent the dataset is useful for understanding the interplay of expressive power and generalization."
>
> R3:
>
> 1. 800 graphs are not small for expressiveness. Recalling that all 800 graphs are non-isomorphic. Thus, generating 319,600 pairs of graphs (from CSL and SR25 settings) or even more by adding noisy components (from EXP settings) will be easy. However, we limit the number to 400 pairs (each graph appears only once) to keep the most essential and difficult comparisons. For more details regarding dataset size, please refer to General Response 2.
> 2. We do not study generalization in this work while focusing purely on expressivity. Generalization can be measured by real-world performance. And we can compare methods with different expressivity and real-world performance to understand the interplay. Generally, higher expressive power leads to better real-world performance. For example, numerous subgraph-based models show remarkable performance with better expressivity (expressivity results in Section 5 subgraph-based models; for real-world performance please see [1] section 10 Experiments); A widely used molecular dataset ZINC [2] owns two SOTA models [3, 4] both focusing on expressiveness. However, in some situations, better theoretical expressiveness does not lead to better performance. Our dataset can exactly help us understand this. For example, PPGN is theoretically as expressive as 3-WL but does not give a promising real-world performance. Our tests show that it actually cannot reach its theoretical expressive power, which can help explain its subpar real-world performance.

---

> > ### Author Response · Authors · 2023-08-21
> > **Response to Reviewer Zvmi [2/2]**
> >
> > >OFI4: "The authors should make it clear what using the dataset gives a user over just theoretically analyzing the expressive power."
> >
> > R4:
> > 1. BREC is not to analyze expressive power theoretically. Theoretically analyzing the expressive power is done in many theoretical proofs and analyses. However, whether these models' practical implementations can achieve their theoretical power is constantly questioned and worth studying. Previous expressiveness datasets only served as a simple and incomplete verification, but BREC is designed to test whether the theoretically powerful models can reach their expressiveness in practice.
> > 2. As we discussed in Section 2, previous datasets face the problem of difficulty, granularity and scale. With the fast development of more expressive GNN variants, a new expressiveness dataset with higher difficulty, finer granularity and larger scale is urgently needed, which is why we propose BREC.
> > 3. BREC is not limited to an upgraded version of previous datasets but provides tons of additional perspectives in expressive power analysis. It can help us better understand each model's practical advantages and limitations through the lens of difficult graphs. For example, some practical choices have the same theoretical expressiveness, like node labeling and distance encoding for subgraph-based models [1]. However, our experiments show that distance encoding leads to higher practically achieved expressiveness, which can inspire new research.
> >
> > >OFI5: "The need for the theoretical framework is not clear. That is, is it really the case that previous evaluations suffer from numerical precision problems?"
> >
> > R5: For previous datasets (EXP, SR25, CSL), numerical precision may not be a problem because they are all in classification settings with few labels. However, when the distribution and number of graphs increase, numerical precision becomes the problem. If we directly use a traditional classification setting, the dataset would turn into an 800-class classification problem (in previous ones, it would be a 2 or 10 or 15-classification problem), where factors other than expressiveness play a more important role. Another previously used evaluation metric is calculating the output distance and determining whether the model successfully distinguishes a pair of graphs by comparing the distance to a manually decided threshold. For CSL and SR25, all the graphs have the same size, i.e., the output is under the same scale. However, graph size varies a lot in BREC. Therefore, the difference between different graphs and fluctuations within the same graphs may overlap. Our core idea is to bridge them to calculate a more reliable threshold.
> >
> > Reference:
> >
> > [1] Zhang et al. A Complete Expressiveness Hierarchy for Subgraph GNNs via Subgraph Weisfeiler-Lehman Tests http://arxiv.org/abs/2302.07090
> >
> > [2] Dwivedi et al. Benchmarking Graph Neural Networks https://arxiv.org/abs/2003.00982
> >
> > [3] Ma et al. Graph Inductive Biases in Transformers without Message Passing http://arxiv.org/abs/2305.17589
> >
> > [4] Feng et al. Towards Arbitrarily Expressive GNNs in O(n2) Space by Rethinking Folklore Weisfeiler-Lehman https://arxiv.org/abs/2306.03266

---

### Author Response · Authors · 2023-08-21
**General Response**

>Question1: “The code may still be hard to start a quick test on models” “It is beneficial to use standard APIs”

Reply1:
We distributed BREC to Pypi https://pypi.org/project/brec/, where researchers can conduct a simple test on their models. Researchers can also yield more detailed and advanced customized usage, referring to https://github.com/GraphPKU/BREC

>Question2: “Current dataset is still small” “400 pairs can be obtained with only 29 different graphs”

Reply2:
1. 800 graphs are not small for expressiveness. Recalling that all 800 graphs are non-isomorphic. Thus, generating 319,600 pairs of graphs (from CSL and SR25 settings) or even more by adding noisy components (from EXP settings) will be easy. However, we limit the number to 400 pairs (each graph appears only once) to keep the most essential and difficult comparisons.
2. Unlike large-scale real-world datasets, which require as many graphs as possible splitted into train/val/test for testing generalization and scalability, our datasets aim to test expressiveness, where we do not need the model to generalize by training on big data. Instead, expressiveness datasets purely aim to measure models’ distinguishing power. Furthermore, we provide a reliable and precise evaluation method to obtain the exact distinguishable number of graphs with nearly **no variance**. This indicates that the current number of graphs is **sufficient** to reflect the expressiveness gap between different methods.
3. Some types of graphs are sampled from exhaustively searching until reaching a certain number, and it is possible to enlarge the datasets by seamlessly sampling more graphs. However, it only applies to Basic and Extension types (some other graphs like CFI or distance regular graphs are too rare to get more) and can lead to an unbalanced distribution (focusing too much attention on simple graphs). Therefore, we only keep 800 graphs.

---

### Decision · Program_Chairs · 2023-09-22

**Decision:**

Reject

**Comment:**

This paper introduces a benchmark dataset designed to assess the expressive capabilities of GNNs and their enhanced variants through empirical analysis. The dataset, named BREC, comprises 400 pairs of non-isomorphic graphs categorized into four distinct difficulty levels, as measured by their distinguishability with the k-WL algorithm.  After the rebuttal and the discussion period, the opinions regarding the merit of this work are still divergent. We had further discussions among area chair, senior area chair and PC chairs. We finally agree that there are some concerns which need to be addressed before publication as follows.  The dataset suffers from small size which may not lead to the convincing conclusion for deep learning methods. Moreover, the usage of the dataset both for theoretical analysis and practical usage needs further justifications. The necessity of the theoretical framework remains ambiguous. Specifically, it's uncertain whether previous evaluations encountered issues related to numerical precision.